



# Using feature-based verification methods to explore the spatial and temporal characteristics of forecasts of the 2019 Chlorophyll-*a* bloom season over the European North-West Shelf

Marion Mittermaier[1], Rachel North[1], Jan Maksymczuk[2], Christine Pequignet[2], David Ford[2]

[1]Verification, Impacts and Post-Processing, Weather Science, Met Office, Exeter, EX1 3PB, United Kingdom
[2]Ocean Forecasting Research & Development, Weather Science, Met Office, Exeter, EX1 3PB, United Kingdom

*Correspondence to*: Marion Mittermaier (marion.mittermaier@metoffice.gov.uk)

## Abstract.

A feature-based verification method, commonly used for atmospheric model applications, has been applied to Chlorophyll-*a* (Chl-*a*) concentration forecasts from the Met Office Atlantic Margin Model at 7 km resolution (AMM7) North West European Shelf Seas model, and compared against gridded satellite observations of Chl-*a* concentration from the Copernicus Marine Environmental Monitoring Service (CMEMS) catalogue. A significant concentration bias was found between the model and observations. Two variants of quantile mapping were used to mitigate against the impact of this bias on feature identification (determined by threshold exceedance). Forecast and observed Chl-*a* objects for the 2019 bloom season (March 1 to 31 July), were analysed, firstly in space only, and secondly as space-time objects, incorporating concepts of onset, duration and demise. It was found that forecast objects tend to be too large spatially, with lower object numbers produced by the forecasts compared to those observed. Based on an analysis of the space-time objects the onset of Chl-*a* blooming episodes at the start of the season is almost a month too late in the forecasts, whilst several forecast blooms did not materialise in the observations. Whilst the model does produce blooms in the right places, they may not be at the right time. There was very little variation in forecasts and results as a function of lead time. A pre-operational AMM7 analysis, which assimilates Chl-*a* concentrations was also assessed, and found to behave more like the observations, suggesting that forecasts driven from these analyses could improve both timing errors and the bias.



# 1 Introduction

The advancements in atmospheric numerical weather prediction (NWP) such as the improvements in model resolution began to expose the relative weaknesses in so-called traditional verification scores (such as the root-mean-squared-error for example), which rely on the precise matching in space and time of the forecast to a suitable observation. These metrics and measures no longer provided adequate information to quantify forecast performance (e.g. Mass et al. 2002). One key characteristic of high-resolution forecasts is the apparent detail they provide, but this detail may not be in the right place at the right time, a phenomenon referred to as the "double penalty effect" (Rossa et al. 2008). This realisation created the need within the atmospheric community for creating more informative yet robust verification methods. As a result, a multitude of so-called "spatial" verification methods were developed, which essentially provide a number of ways for accounting for the characteristics of high-resolution forecasts.

In 2007 a spatial verification method inter-comparison (Gilleland et al. 2009, 2010) was established with the aim of providing a better collective understanding of what each of the new methods was designed for, diagnosing and categorising what type of forecast errors each could quantify. A decade later Dorninger et al. (2018) revisited this inter-comparison, adding a fifth category so that all spatial methods fall into one of the following groupings: neighbourhood, scale separation, feature-based, distance metrics and field deformation.

The use of spatial verification methods has therefore become commonplace for atmospheric NWP (see Dorninger et al. 2018 and references within). Neighbourhood-based methods in particular have become popular due to the relative ease of computation and intuitive interpretation. Recently one such neighbourhood spatial method was demonstrated as an effective approach for exploring the benefit of higher resolution ocean forecasts (Crocker et al. 2020). Another class of methods focus on how well particular features of interest are being forecast. Forecasting specific features of interest is one of the main reasons for higher horizontal resolution. Feature-based verification methods, such as the Method for Object-based Diagnostic Evaluation (MODE, Davis et al. 2006) and the time domain version




MODE-TD (Clark et al. 2014) enable an assessment of such features, focusing on the physical attributes
of the features (identified using a threshold) and how they behave at a given point in time, and evolve
over time. These methods require a gridded truth to compare to. Whilst the initial inter-comparison
project was based on analysing precipitation forecasts, over recent years their use has extended to other
variables, provided gridded data sets exist that can be used to compare against (e.g. Crocker and
Mittermaier 2013, Mittermaier et al. 2016). Mittermaier and Bullock (2013) detailed the first of the
MODE-TD prototype tools to analyse the evolution of cloud breaks over the UK using satellite-derived
cloud analyses.

In the ocean, several processes have strong visual signatures that can be detected by satellite sensors.
For example, mesoscale eddies can be detected from sea surface temperature or sea level anomaly (e.g.
Chelton et al. 2011, Morrow and Le Traon, 2012, Hausmann and Czaja, 2012). Phytoplankton blooms
are seasonal events which see rapid phytoplankton growth as a result of changing ocean mixing,
temperature and light conditions (Sverdrup, 1953, Winder and Cloern, 2010, Chiswell, 2011).  Blooms
represent an important contribution to the oceanic primary production that is a key process for the
oceanic carbon cycle (Falkowski et al. 1998). Their spatial extent and intensity in the upper ocean make
them visible from space with ocean colour sensors (Gordon et al. 1983, Behrenfeld et al. 2005).
Biogeochemical models coupled to physical models of the ocean provide simulations for the various
parameters that characterise the evolution of a spring bloom. In particular, Chlorophyll-*a* (Chl-*a*)
concentrations provide an index of phytoplankton biomass. Chl-*a* concentration can also be estimated
from spaceborne ocean colour sensors (Antoine et al. 1996).

Validation of marine biogeochemical models has traditionally relied on simple statistical comparisons
with observation products, often limited to visual inspections (Stow et al. 2009; Hipsey et al. 2020). In
response to this, various papers have outlined and advocated using a hierarchy of statistical techniques
(Allen et al. 2007a, b; Stow et al. 2009; Hipsey et al. 2020), multivariate approaches (Allen and
Somerfield, 2009), and novel diagrams (Jolliff et al. 2009). Many of these rely on matching to
observations in space and time, but some studies have started applying feature-based verification





methods. Emergent properties have been assessed in terms of geographical provinces (Vichi et al. 2011)
and ecosystem functions (de Mora et al. 2016). In a previous application of spatial verification methods
developed for NWP, Saux Picart et al. (2012) used a wavelet-based method to compare Chl-*a*
concentrations from a model of the European North West Shelf to an ocean colour product.

For this paper, both MODE and MODE-TD (or MTD for short) were applied to the Met Office Atlantic
Margin Model at 7 km resolution (O'Dea et al. 2012, Edwards et al. 2012, O'Dea et al. 2017, King et al.
2018) for the European North West Shelf (NWS), in order to evaluate the spatio-temporal evolution of
the bloom season in both forecast and observation fields.

In Section 2 the data sets used in the verification process are introduced. Section 3 describes MODE and
MTD. Section 4 contains a selection of results, and their interpretation. Conclusions and
recommendations follow in Section 5.
**2 Data sets for the 2019 Chl-*a* bloom**
As stated in Section 1, feature-based methods such as MODE and MTD require a gridded field of some
description. In order to assess the European NWS Chl-*a* concentration forecast (AMM7v8), a satellite-
based gridded ocean colour product (L4) product and model assimilative analysis (AMM7v11) are
considered as gridded "truth" sources.
**2.1 Satellite-based gridded ocean colour products**
A cloud-free gridded (space-time interpolated, L4) daily product delivered through the Copernicus
Marine Environment Monitoring Service (CMEMS) catalogue provides Chl-*a* concentration at ~1 km
resolution over the Atlantic (46°W–13°E, 20°N–66°N). The L4 Chl-*a* product is derived from merging
of data from multiple sensors: MODIS-Aqua, VIIRSN and OLCI-S3A. The near-real-time (NRT)
products, which are computed one day after satellite acquisition, were downloaded after a few days to
benefit from the delayed-time (DT) update that provides a better-quality product. The satellite derived
estimate is an integrated value over optical depth.






Errors in satellite-derived Chl-*a* can be more than 100% of the observed value (e.g. Moore et al., 2009).
The errors in the L4 Chl-*a* values are often at their largest near the coast, especially near river outflows.
However, in the rest of the domain, smaller values of Chl-*a* mean that even large percentage
observation errors result in errors typically smaller than the difference between model and observations.
As will be shown, the models at 7 km resolution cannot resolve the coasts in the same way as is seen in
the satellite product.

For this study the ~1 km resolution L4 satellite product was interpolated onto the AMM7 grid using
standard two-dimensional horizontal cubic interpolation. This coarsening process retained some of the
larger concentrations present in the L4 product.

**2.2 Model forecasts and analyses**
**2.2.1. Forecasts**
Forecasts of ocean physics and biogeochemistry for the European NWS waters are delivered through
CMEMS. For a summary of the principles underlying the service see e.g. Le Traon et al. (2019).

The hydrodynamics of the NWS is provided by the Forecasting Ocean Assimilation Models (FOAM)
system which consists of a NEMO-based (Nucleus for European Modelling of the Ocean, Madec et al.
2016) hydrodynamic model coupled to the variational data assimilation scheme (NEMOVar – Waters et
al., 2015, King et al., 2018, O'Dea et al 2017).  For the NWS region, FOAM is configured for the
shallow water of the shelf sea. Coupled to FOAM is the European Regional Seas Ecosystem Model
(ERSEM) which provides forecasts for the lower trophic levels of the marine food web (Butenschön et
al. 2016). Satellite and in situ sea surface temperature (SST) observations are assimilated using a 3D-
Var method (King et al., 2018). The forecasts run on the Atlantic Margin Model grid at approximately 7
km horizontal resolution (AMM7) from 40 °N, 20 °W to 65 °N, 13 °E. Daily mean Chl-*a* concentration
forecast out to Day 4 for the period of 1 March-31 July 2019 were compiled from the current





operational version (hereafter referred to as AMM7v8). Note that the analysis (1-day hindcast) and
forecasts used here are available from the CMEMS catalogue.

Ideally, Chl-*a* concentration from the model should be integrated over optical depth to be equivalent to
the satellite derived value defined in 2.1 (Dutkiewicz et al. 2018). However, this is currently a non-
trivial exercise, and cannot be accurately calculated from offline outputs. Therefore, the commonly
accepted practice is to use the model surface Chl-*a* (Lorenzen 1970, Shutler et al. 2011). Here it is
assumed that the difference between surface and optical depth-integrated Chl-*a* is likely to be small in
comparison with the actual model errors.

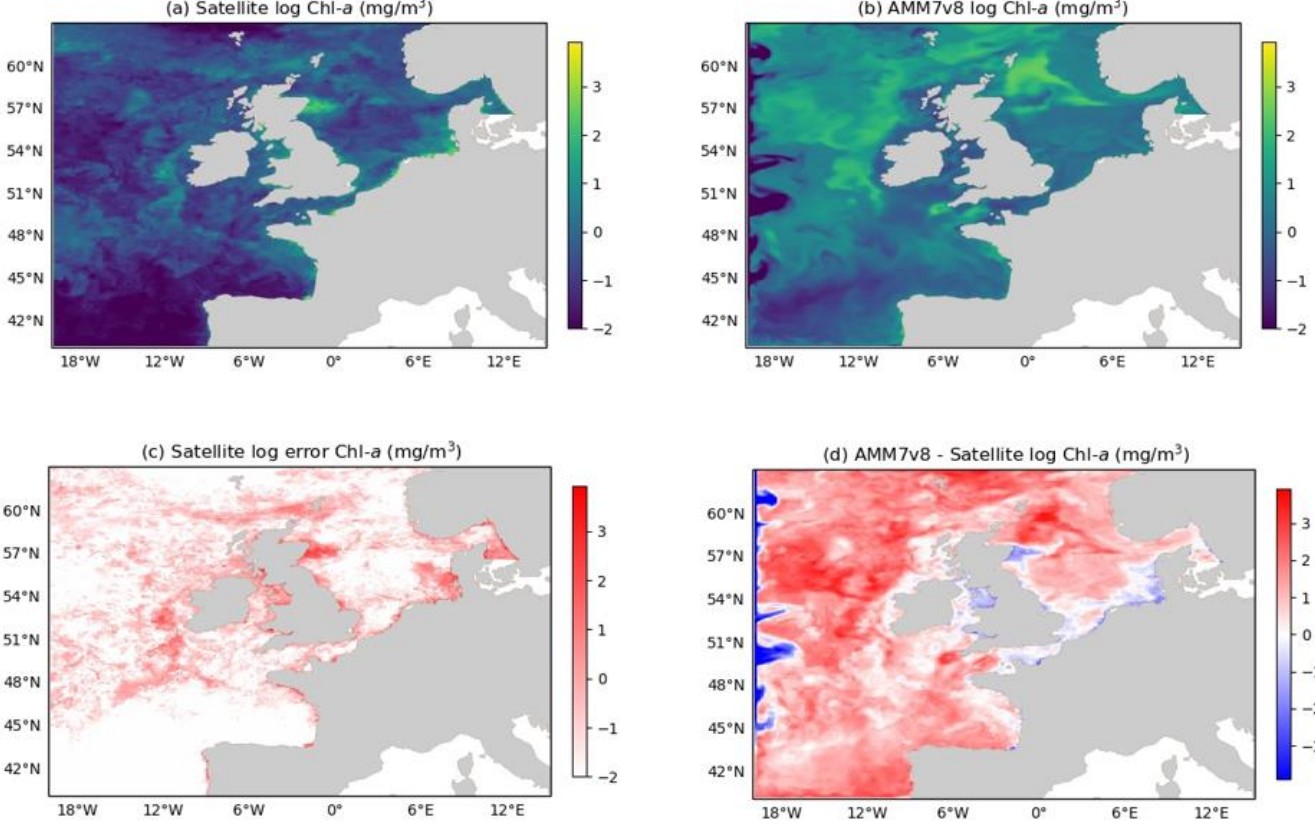


**Figure 1. (a) Daily mean L4 multi-sensor observations (top left) regridded on the 7km resolution model grid and (b) AMM7v8**
**output (top right) Chl-*a* for 1 June 2019. (c) Error estimates on the multi-sensor L4 Chl-*a* (bottom left) and (d) difference between**
**model and observations (bottom right)**





Figure 1 shows the L4 ocean colour product (left) and AMM7v8 analysis (right) for 1 June 2019 on the
top row, using the same plotting ranges. The second row shows the difference field that is provided with
the L4 ocean colour product (left), and the AMM7v8 minus L4 difference field (right). The mean error
(bias) is generally positive with the AMM7v8 analysis containing higher Chl-*a* concentrations,
especially in the deeper North Atlantic waters. The exceptions are along the coast where the AMM7v8
analysis is deficient, but it should be noted that these are also the zones where some of the largest
satellite retrieval errors occur and where a 7-km resolution model, with a coarse representation of the
coast, does not fully represent complex coastal and estuarine processes. The differences between the
analysis and the L4 product can be comparable in size to the retrieval errors.

**2.2.2. Analyses**
As well as the products from the CMEMS catalogue detailed above, there was also an opportunity to
use model analyses provided from the latest pre-operational AMM7-ERSEM model due for release in
late 2020 – hereafter referred to as AMM7v11. This system incorporates upgraded physics, and an
improved data assimilation scheme including additional observations. Specifically, in addition to
assimilation of more physical variables (water column temperature and salinity profiles and sea level
anomaly), this new version includes assimilation of satellite-borne ocean colour Chl-*a* concentrations.
The satellite ocean colour observations assimilated are from a daily multi-sensor composite product
based on MODIS and VIIRS) with resolutions of 1 km for the Atlantic (for further information see
OCEANCOLOUR_ATL_CHL_L3_NRT_OBSERVATIONS_009_036 on the CMEMS catalogue).

Significant differences between the AMM7v11 and AMM7v8 (the forecast version) relevant to the
biogeochemistry include new coupling through the Framework for Aquatic Biogeochemical Models
(FABM, Bruggeman and Bolding, 2014), an improved river discharge dataset and new nitrogen
deposition input. Note only the analysis (Day 0) of AMM7v11 (i.e. no corresponding forecasts) was
available at the time of the assessment.





## 3 Method for Object-based Diagnostic Evaluation (MODE) and MODE Time-Domain (MTD)

**3.1. Description of the methods**

This section provides a description of the Method for Object-Based Diagnostic Evaluation (MODE) tool, first described in Davis et al. (2006) and its extension MODE Time-Domain (MTD).

MODE and MTD can be used on any sequence of forecasts which contain a feature that is of interest to a user (whoever that user may be, model developer or more applied), thus mimicking what humans do. Therefore, they can be used in a very generalised way, comparing two fields: in this context one is a forecast, the other an observation-based gridded field or model-based analysis. MODE identifies the features (called objects), as areas for which Chl-*a* concentrations values exceed a threshold, in both the forecast and observed fields. Object attributes are calculated and compared. Simple objects can be *merged* (to form clusters) within a single forecast or observed field and *matched* to objects in the other field. Summary statistics describing the objects and object pairs are produced. These statistics can be used to identify similarities and differences between forecast and observed objects, which can provide diagnostic insights of forecast strengths and weaknesses.

Briefly, applying MODE consists of the following steps (which are described in detail in Davis et al. 2006):

1) Both forecast and observation (or analysis) need to be on the same grid. Typically, this means interpolating the observations to the model grid to avoid the model being expected to resolve features which are sub-grid scale.

2) Depending on how noisy the fields are they need to be smoothed *further*. Here convolution is used as the method and is based on a disk. The choice of smoothing (convolution) radius depends on the field to be evaluated. It is worth remembering that the numerical discretisation implies that any model's true resolution (i.e. the scales which the model is resolving) is between 2 and 4 times the horizontal grid (mesh) resolution. The number of areas identified will vary inversely with the convolving radius.

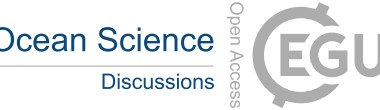

3) Define a threshold which captures the feature of interest and apply it to both the smoothed forecast and observed fields to identify simple objects.

4) The original intensity information in the field is then reinstated in the identified features (i.e. the analysis of the object attributes is *not* based on the smoothed fields).

5) Depending on the merging option that is chosen, simple objects that are identified as being related to each other are merged to form cluster (complex) objects.

6) Lastly, objects in the forecast and observed fields can be matched based on a range of criteria using a fuzzy logic engine (low level artificial intelligence), which together are expressed as the so-called "interest" score. The higher the score the stronger the match. All objects are compared in both fields and interest scores are computed for all. A threshold is set on the interest score value (typically 0.7) to denote which are the best matches to provide a unique best match for each object pair. Some objects will remain unmatched (either because there is none or because there are no interest values high enough to provide a credible match) and these can be analysed separately.

Simple forecast and observed (analysis) object attributes which can be evaluated include *centroid location, area, axis angle, curvature and aspect ratio*. They can also be split into matched and unmatched to see what proportion of objects are matched, for instance. Matched object pairs have different attributes such as *centroid difference, angle difference, union* and *intersection area* for example, focusing on the comparison between the matched objects in terms of how far apart they are, whether they are the same size etc.

From the above it is clear that MODE is highly configurable. To gain an optimal combination of configurable parameters for each application requires extensive sensitivity testing to gain sufficient understanding of the behaviour of the data sets to be examined, and to achieve, on average, heuristically the right outcome. Initial tuning requires user input to check whether the method is replicating what a human would do.





1) The sensitivity to threshold and smoothing (convolution) radius should be explored. Numerically information such as the object counts, and areas associated with each combination of threshold and smoothing radius can be summarised into what is known as a "quilt plot".

2) The sensitivity to the merging option must also be investigated. The options provided include *none, threshold only* (using double thresholds), a *fuzzy logic engine*, or a *combination of both threshold and fuzzy logic*. Depending on the field this could have an impact. In this instance the merging option had very little impact.

3) The behaviour of the matching can also be configured. The interest values that are computed for each possible pair of forecast-observation objects are thresholded to define which objects match. Options include *no_merge*, *merge_forecast* and *merge_both*. There is an increase in computation expense for the *merge_both* option, which may, or may not, be necessary for a given application.

Note also that a minimum size (area) is set for object identification. This is often a somewhat pragmatic choice. If the size is set too small, too many objects are identified, which end up being merged. If too large, very few objects are identified. In this study the *merge_both* option was used for MODE with a minimum area of 10 grid squares (~70 km$^2$).

Identical to MODE, identifying time-space objects in MTD uses smoothing and thresholding. Applying a threshold yields a binary field where grid points exceeding the threshold are set to one. At this stage each contained region of non-zero grid points in space and time is considered a separate object, and the grid points within each object are assigned a unique object identifier. For MTD the search for contiguous grid points not only means examining adjacent grid points in space, but also the grid points in the same or similar location at adjacent times to define a space-time object. The same fuzzy logic-based algorithms used for merging and matching in MODE apply to MTD as well. Similarly, to MODE a minimum volume of 1000 grid squares was imposed for space-time object identification. For MTD a lower interest score of 0.5 is used for matching objects.



MODE and MTD produces object attributes for both "single" and "paired" objects (when matching), as
well as for "simple" or "cluster" (when merging objects within either the forecast or analysis field)
object attributes. Throughout this analysis the *single simple* objects have been used when considering
forecast-only or analysis-only attributes.

**3.2 Defining Chl-*a* concentration thresholds and other choices on tuneable parameters**


Chl-*a* can vary over several orders of magnitude. Often $\log_{10}$ thresholds are used to match the fact that
Chl-a follows a lognormal distribution (e.g. Campbell 1995). Defining thresholds can be difficult: on
the one hand there is the desire to capture events of interest, so the thresholds should not be too low,
whereas on the other hand if the thresholds are too high no events are captured and there is nothing to
analyse. From a regional perspective the values of interest are in the range of 3–5 mg.m$^{-3}$ (Schalles,
2006). For this study a range of $\log_{10}$ thresholds between 0.2 and 1.4 mg.m$^{-3}$ were defined,
corresponding to a lowest threshold of 1.62 and a highest threshold of 25 mg.m$^{-3}$.

In addition to the interpolation of the L4 ocean colour product onto the AMM7 grid a smoothing radius
of 5 grid points was also applied to the observed fields to remove some of the very small and noisy
objects typically found near the coast (which neither AMM7v8 nor AMM7v11 can resolve). No
smoothing was applied to the forecasts or model analyses as these were considered to be smooth
enough. This radius was identified based on the sensitivity analysis, which will be described in more
detail in Section 4. This sensitivity analysis also identified the concentration thresholds which were
viable for analysis. Only the 2.5 mg.m$^{-3}$ threshold will be discussed here. For this study the default
settings in MODE were used to compute the interest score.

**3.3 Software used**

Verification was performed using the Model Evaluation Tools (MET) verification package that was
developed by the National Center for Atmospheric Research (NCAR), and which can be configured to
generate both MODE and MTD outputs. MET is freely available for download from GitHub at





https://github.com/dtcenter/MET. For this study version 8.1 of the software was used. MET allows for a
variety of input file formats but some pre-processing of the CMEMS NetCDF files was necessary
before the MODE package could be applied. This includes regridding of the observations onto the
model grid, and addition of forecast lead time and forecast reference time variables to the NetCDF
attributes. All these attributes are detailed in the MET software documentation (Newman et al. 2018).
**4. Data analysis**
The data analysis presented in this section focuses on a subset of results computed for the following:
• Comparing the L4 ocean colour product to the AMM7v8 and AMM7v11 analyses
• Comparing the AMM7v8 forecasts to the L4 and AMM7v11 analyses

**4.1 Understanding concentration differences and associated impacts**

Figure 1 suggests a considerable bias between the AMM7v8 forecasts and the L4 ocean colour product.
Whenever a threshold is applied to define the range or features of interest, the presence of a bias can
render the results impossible to interpret because being a spatial method, the object area forms an
important part of any comparison. Consider for example the case where the bias is such that whilst
features are present, they are so different in magnitude that the objects can only be identified in one of
the fields, and not the other. Whilst it could be useful to simply analyse the unmatched objects, the
purpose of MODE is to consider whether features are forecast correctly and if there are no matched
pairs then this is impossible to do.

This is illustrated in Figure 2 which shows the daily Chl-$a$ concentrations as represented in L4, and the
AMM7v8 and AMM7v11 analyses. The raw fields are plotted in (a) to (c). The AMM7v8 analysis in (a)
is markedly different to (b) and (c). Applying a threshold of 6.3 mg.m$^{-3}$ yields 12 objects in the
AMM7v8 analysis, none in the AMM7v11 analysis and 6 in the L4 product. If these options were
verified against each other, some comparisons would yield no matched pairs. If the objective is to see if
the forecast has any skill in forecasting features (not just absolute concentrations) that data analysis


would yield no useful information. In that case the most sensible thing to do is to provide some form of
bias removal to mitigate against the impact of the concentration differences affecting the ability to
understand whether, at a base level, the forecasts have any skill at forecasting the features (blooms).

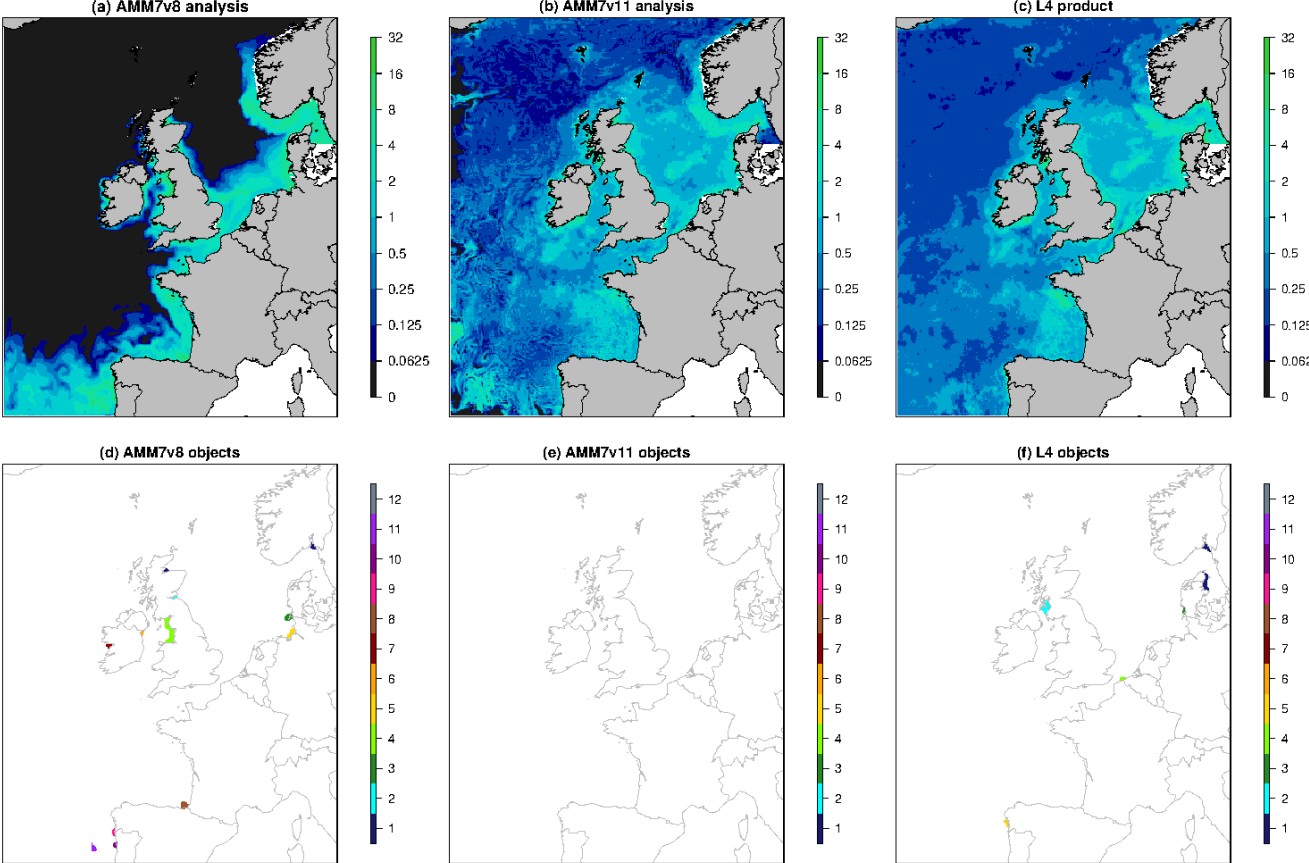


**Figure 2: Daily Chl-*a* concentrations (in mg.m$^{-3}$) for 29 March 2019 showing the three different analyses in (a) to (c). If a constant**
**threshold of 6.3 mg.m$^{-3}$ is applied then MODE finds 12 objects exceeding this threshold (d), where the colour matches the object**
**number.  No objects are identified in AMM7v11 (e) and 6 in the L4 ocean colour product (f). The raw fields in (a) to (c) indicate a**
**considerable difference in concentrations between the analyses with AMM7v11 much closer to the L4 ocean colour product. The**
**AMM7v8 analysis is indicative of the AMM7v8 forecast behaviour too.**

To understand the nature of the concentration differences better the study data set was turned into
cumulative distribution functions (CDF) of the $\log_{10}$ Chl-*a* concentrations, by taking all grid points in
the domain and all dates in the study period. This was done for the L4 ocean colour product and the





AMM7v8 analyses, the two that clearly differ more dramatically from Fig 2. These are plotted in Fig. 3,
showing that the differences are not just due to an offset in the concentrations but a more complex
difference. Close to half of the AMM7v8 analyses concentrations are significantly lower than observed,
some extremely low (at the numerical noise level), whilst the L4 distribution's smallest concentrations
are several orders of magnitude greater. The two distributions cross over around ~3 mg.m$^{-3}$, and whilst
the shape of the upper half of the AMM7v8 and L4 CDFs shows the same rate of increase, here the
AMM7v8 values are now larger than the L4 values. The L4 concentrations span a much smaller range
in magnitudes providing a much tighter distribution with approximately 95% of the values below
concentrations of 10 mg.m$^{-3}$. Generally, the AMM7v8 does not contain as many larger concentrations
so that the peak concentrations are too low when compared to the L4 product. The shape of this
distribution shows that a bulk bias correction scheme which relies on a simple addition or subtraction
(because the distributions are shifted) would not work. This situation requires a method like quantile
mapping, which preserves the shape of the distribution.

In practice the application of a quantile mapping method means that the threshold-exceedance seen in
the forecasts occurs at the same proportion as that seen in the observations. This frequency equivalence,
applied across the whole field, behaves as a bias removal tool. To explain quantile mapping another
way, the observed values at that time are ranked and the threshold value is determined as a quantile of
that distribution. The equivalent quantile is then selected from the ranked forecast values.

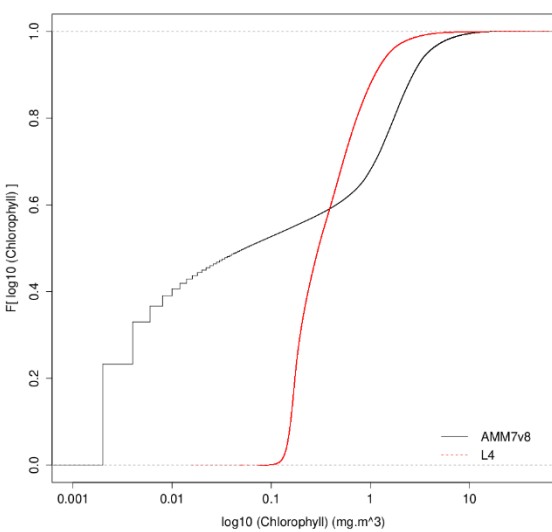

**Figure 3. Empirical cumulative distribution functions of the log10 Chl-*a* concentration for observations (L4 ocean colour product) and Day 1 forecasts from AMM7v8 for the 2019 bloom season.**

It is probably too simplistic to call the differences a bias, but the impact the documented concentration differences may have on identifying objects (through the use of fixed concentration thresholds) for the purposes of analysing object properties (which almost exclusively relate to the spatial properties of the fields), needed to be minimised.

For the analysis that follows quantile mapping was applied in one of two different ways, necessitated by what functionality was available in the MET software. For the 2-dimensional MODE analysis the option to remove the bias can be specified (available from MET v8.1) which performs a quantile mapping between the two fields for each forecast-analysis pair. Here the observed threshold is specified (fixed) and a ranking of values in both the forecast and observed field identifies the analysis value that has the equivalent rank in the forecast distribution. In this instance the forecast threshold varies with time to ensure that the frequency bias of the paired fields is equal to one at all times.

For the three-dimensional MTD analysis tool this option was not available as yet. In this instance the seasonal distribution shown in Fig. 3 was used to derive a seasonal threshold denoting a percentile





equivalence across the two datasets. The reference (fixed) threshold is based on the L4 product. In this
instance the day-to-day frequency bias will not necessarily be 1 but the frequency bias will be
approximately 1 when the season is taken as a whole.

Once the bias has been taken account of in this manner, the spatial properties of the subsequent
identified objects can be analysed without the concern that the concentration differences are leading to a
misinterpretation of results (remembering that the primary purpose of a feature-based assessment is to
determine whether features of interest can be identified with any skill).

From Fig. 2 the concentration differences between the AMM7v11 analysis and L4 ocean colour product
seem to be much reduced. MODE was used to compare these two "truths" by treating the AMM7v11
analysis as the 'forecast' field with the latter as an observation field to understand what the day-to-day
differences in thresholds are. Figure 4 provides the time series of AMM7v11 thresholds which provide
the quantile (frequency) equivalence to 2.5 mg.m$^{-3}$. There are still differences in behaviour between the
two sources, but especially early on in the season the differences are small. Larger day-to-day variations
are evident as the season progressed, where the threshold cycles between values of ~2.5 mg.m$^{-3}$ and ~4-
5 mg.m$^{-3}$. There are notable peaks at the end of May and the beginning of July. At these times the
AMM7v11 appears to have higher Chl-$a$ concentrations in large portions of the domain compared to the
L4 product. The AMM7v11 threshold for the season is 2.9 mg.m$^{-3}$, which can be considered a relatively
small variation. From this result it would seem that the satellite observations constrain the model initial
conditions, both in terms of the minimum values and also limiting any tendency to bloom where it is not
seen in observations. The lack of constraint is very apparent in AMM7v8, as shown in Fig 5.





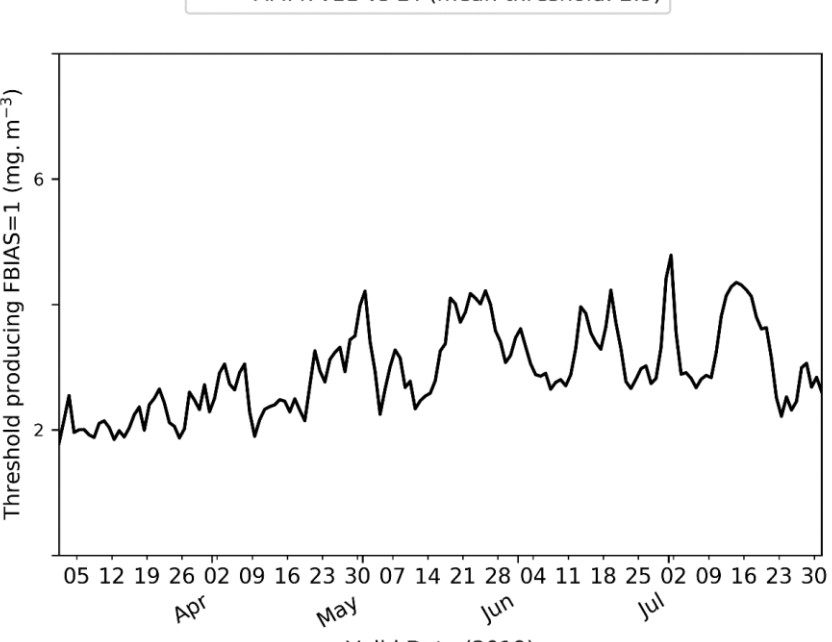

**Figure 4. Threshold identified as producing a frequency bias of 1 for the AMM7v11 analysis compared to L4 ocean colour product. The mean threshold over the 2019 season, 2.9 mg.m$^{-3}$, is indicated in the legend. The observed threshold used was 2.5 mg.m$^{-3}$.**

**Error! Reference source not found.** illustrates the AMM7v8 threshold variations based on a day 4 forecast compared to AMM7v11 and L4 across the 2019 bloom season using the built-in functionality in MODE as for Fig 4. The same threshold of 2.5 mg.m$^{-3}$ was used for both the AMM7v11 and L4 products, whilst the AMM7v8 forecast thresholds are derived with respect to these two analyses. It is worth noting that there is very little variation with forecast lead time (and will not be shown), hence showing the day 4 forecast values is fairly representative of the AMM7v8 analysis and all forecast lead times.

Within the first month, before the bloom started in earnest, the thresholds are similar to, if not slightly under, the observed value. This is consistent with Fig. 3. The forecast threshold values steadily increase through April, until at the end of the month there is a spike in the threshold required to maintain the



frequency bias at one. Looking at the MODE graphical output for this time period suggests that
AMM7v8 has increased Chl-*a* concentrations in both the Bay of Biscay, the Norwegian Sea and North
Sea which may account for this. From mid-May onwards, as the bloom extends to most of the offshore
regions, the threshold increases most, peaking at ~13 mg.m$^{-3}$. Investigating the objects identified over
this period it can be seen that the forecasts are very active in the South West Approaches and the North
Sea, in addition to north-west of Scotland and in a region off the northern domain edge. The latter
object is not identified in the L4 ocean colour product. The spike towards the end of June coincides with
an area of elevated forecast Chl-*a* in the North Atlantic, between Iceland and the United Kingdom. The
region affected is physically far larger than seen in the observations. By the end of the bloom season,
the threshold values are back down to similar values as the observed threshold. By contrast the forecast
thresholds derived when using the AMM7v11 analysis are smaller but follow the same general pattern –
providing evidence that the AMM7v11 analysis sits somewhere between the L4 and AMM7v8 in terms
of concentrations but is still closer to the L4 product, as shown in Fig 4. The assimilation process
provides a smoothing effect, which also means that peaks seen in the L4 will have been reduced in the
AMM7v11 analysis, for example.

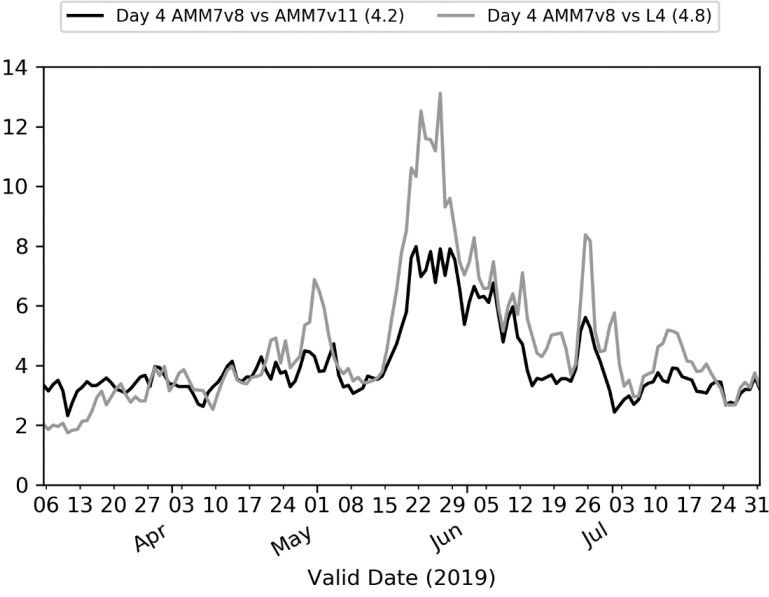


**Figure 5. Forecast threshold value (mg.m$^{-3}$) which produced a frequency bias of 1 for the AMM7v8 day 4 forecasts versus L4**
**satellite product (grey), and for the AMM7v8 day 4 forecasts against the AMM7v11 analysis (black). The L4 and AMM7v11**





threshold used is 2.5 mg. m$^{-3}$. The average value for the threshold across the time series is in brackets in the legend.

For the MTD analysis objects in the L4 ocean colour product and the AMM7v11 analyses were defined using a Chl-*a* concentration threshold of 2.5 mg.m$^{-3}$, whereas for the AMM7v8 forecasts and analysis a threshold of 6 mg.m$^{-3}$ was used, derived from the CDFs plotted in Fig. 3. This is slightly higher than the mean value derived in Figs 4 and 5 showing that the seasonal CDF does provide a slightly different overall range of concentrations than day-to-day variations.

**4.2 Sensitivity analysis**

In order to ensure that MODE used optimal settings for the ocean forecasts under study, the sensitivity of results to smoothing and Chl-*a* concentration were investigated to find the best object identification results, balancing the need for identifying objects with keeping the number of objects manageable.

Much of the initial identification of thresholds and smoothing requirements was done using data from the 2018 bloom season. It is worth noting that this work was done without accounting for the concentration differences but simply analysing the distributions inherent within the data sets. Figure 6 provides a selection of quilt plots derived from using the L4 ocean colour products and AMM7v8 analyses during July 2018, using one of the merging options which was tested. As stated earlier, results for other options were very similar and will not be shown.

The quilt plots essentially provide a two-dimensional mapping of frequencies or counts produced by running MODE multiple times with different settings for the level of smoothing (convolution) radius along the x-axis and increasing concentration thresholds along the y-axis.

In Figure 6 some quilt "difference" plots are shown to focus on the individual characteristics of the AMM7v8 analysis and the L4 ocean colour product based on a set of initial data that was available for July 2018. Here the *merge_both* matching option is shown. In (a) the difference in the number of





simple AMM7v8 and L4 objects is shown as a function of smoothing radius and concentration
threshold. In (b) the difference in median object areas for each combination is shown based on all
objects identified in the July 2018 study period.

From Fig. 6 it is clear there is switch in the sign of the object count "bias" for thresholds above 2.5
mg.m$^{-3}$, where the AMM7v8 analysis has far more objects than the L4 ocean colour product.
Conversely at or below this threshold there are far more L4 objects identified than AMM7v8 objects.
Further examination shows that there are very few L4 objects above 2.5 mg.m$^{-3}$ of any sensible size, so
this was chosen as the threshold for identifying Chl-*a* bloom objects. The median object area increases
with increasing smoothing so that the largest areas occur for the largest smoothing radii. It is therefore
logical that the potential for variations and larger differences increases also with increasing smoothing
radius. This is shown in (b) where it is apparent that the differences between the data sets becomes
larger with increasing smoothing, thus suggesting an upper limit of 6 grid squares on the smoothing
radius for the L4 product. The starkest differences, and hence the need for addressing the concentration
differences before proceeding with any formal analysis is shown in (c). It shows the difference in the
total area enclosed within an object for the data set considered (July 2018). *All* the differences are
positive, i.e. the AMM7v8 object areas in their entirety completely swamp the L4 object areas.

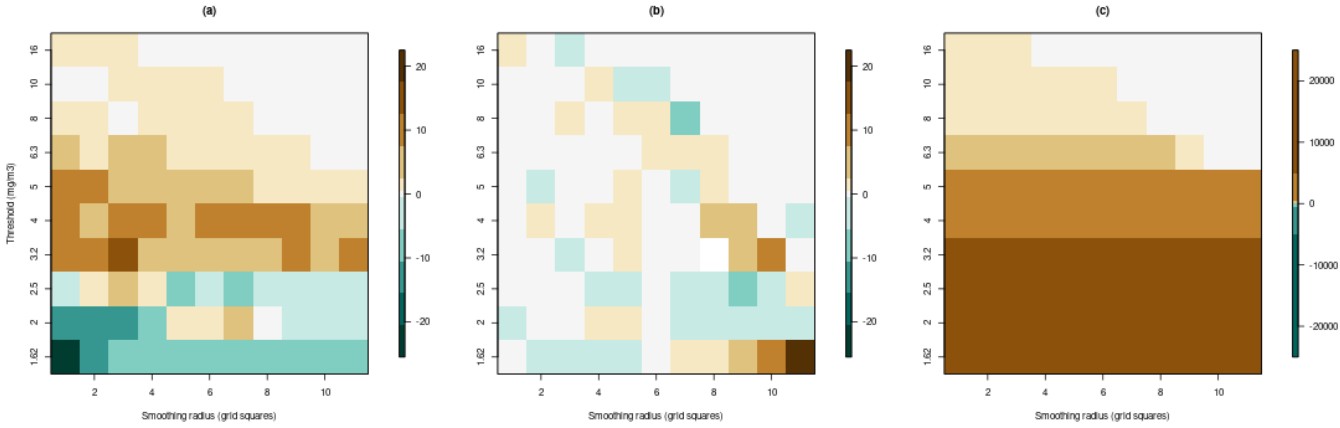



**Figure 6. Quilt "difference" plots for the sensitivity to smoothing (convolution) radius as a function of threshold, showing the**
**difference between AMM7v8 analysis and L4 ocean colour product object (AMM7v8 minus L4): (a) Difference in simple object**



**counts, (b) difference in the median areas (in grid squares over the period), and (c) difference in total area (adding all objects**
**together for each field, also in grid squares). Here the results for the *merge_both* option are shown. Results are for July 2018.**

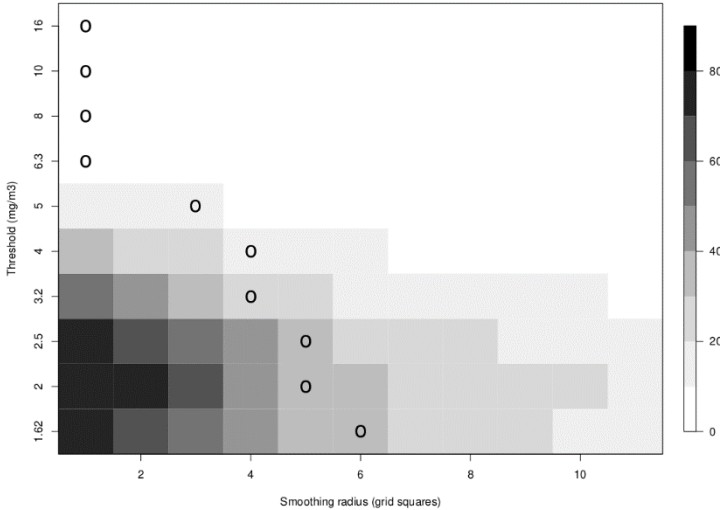

**Figure 7. Average daily object counts for July 2018 produced by adding the L4 and AMM7v8 objects together. Also shown is the**
**smoothing radius which ensures that there are no more than 30 objects (in total) on any given day that have to be analysed. Based**
**on this a smoothing radius of 5 was used for the L4 product and 2.5 mg.m$^{-3}$ threshold.**

The decision on smoothing radius was based on the average daily object count (which is a sum of the observed and forecast object counts). Based on visual inspection it is clear that more than 30 objects become difficult to analyse. This was used as the threshold to examine what the minimum smoothing radius is for each threshold that would ensure that the average daily object count is less than 30. Both these quantities are shown in Fig. 7. This suggests that smoothing needs to be reduced with increasing concentrations because objects become smaller and are less frequent. Too much smoothing could potentially remove these more intense objects from the analysis, though one has to ask the question whether these are genuine and whether meaningful statistics can be compiled if only a few objects are identified. AMM7v8 output is on a ~7 km grid. Given an understanding of what length scales are resolvable in the AMM7 models it was decided that further smoothing of the AMM7 data was not advantageous given the characteristics of the fields at the grid scale. However, it was decided that a





smoothing radius of 5 grid squares (~35 km) for the L4 ocean colour product would be beneficial to
reduce some of the mismatches around the coast.

How similar are the L4 ocean colour product and the AMM7v11 analysis? Put differently, how closely
does the AMM7v11 analysis follow the most important observation source used to produce it? Figure 8
shows the evolution of the proportion of matched object areas (to total area) through the 2019 season,
when using MODE to compare the L4 and AMM7v11 analysis, to further explore the differences (and
similarities) between them. The relatively high levels during April are due to the large numbers of well-
matched, physically small coastal objects in addition to the larger Chl-*a* bloom originating in the Dover
Straits. There is a notable minimum at the beginning of July. Inspecting the MODE graphical output
reveals this is in part due to only a few small objects being identified, and this is compounded by their
complete mismatch; the L4 objects are all coastal, whilst the AMM7v11 objects are either coastal (but
not in the same location as L4 objects) or in the North Atlantic, to the north-west of Scotland. The
relatively high proportions either side of this time arise from a better correspondence in placement of
the coastal objects (there is a distance limit on how far objects can be apart for the matching process to
have a positive contribution to the interest score).



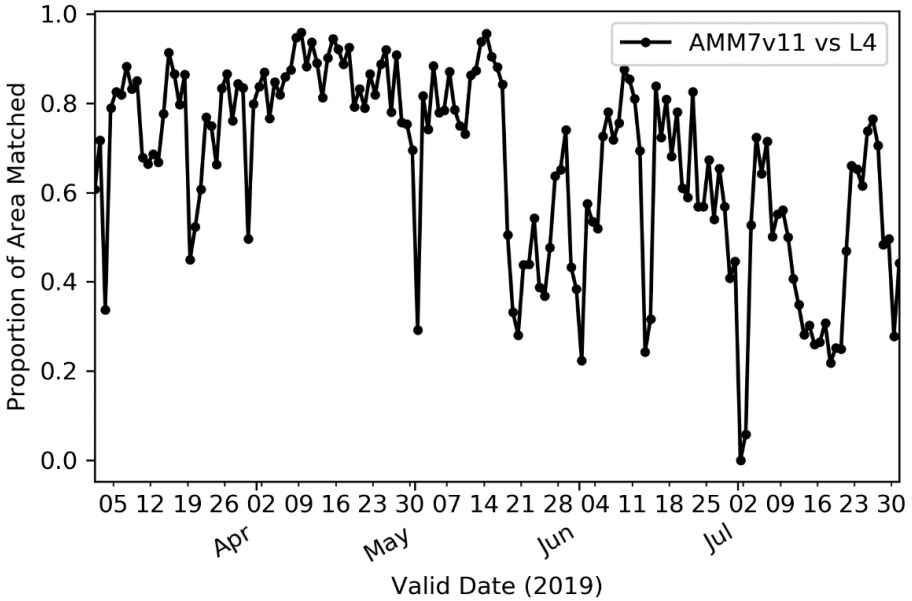

**Figure 8. Proportion of total object area which is matched. Underlying matched and unmatched object areas (in units of numbers of grid squares) are taken from the MODE Analysis output. The threshold used to identify objects is based on the L4 value exceeding 2.5 mg.m$^{-3}$.**

Overall, it will be shown that the AMM7v11 analysis is much closer to the L4 observations than the AMM7v8 analysis. Therefore, the AMM7v11 can be used as a credible source for assessing the AMM7 forecast model system going forward. The AMM7v8 analysis on the other hand, does not resemble the L4 observations sufficiently, and should not be used for assessing the forecasts. The major benefit of using a model analysis is that it is at the same spatial resolution, with the same ability to resolve Chl-*a* bloom objects (i.e. limits the uncertainty due to whether an object could be missing due to the inability of the model to resolve the feature). At this model resolution any coastal objects do not feature in any subsequent data analysis.

Subsequently, results are presented against both the AMM7v11 analysis and the L4 observations to see what effect the truth source may have and whether it could change the assessment of the AMM7v8 forecasts (and analysis).



## 4.3 Examining the MODE object attributes

This section demonstrates the kinds of results that can be extracted from the two-dimensional MODE
objects. Aspects of the marginal (forecast or observed only) and joint (matched/paired) distributions can
be examined. This includes object size (as a proxy for area) but also the proportion of areas that are
matched or unmatched. This part of the analysis in particular is made possible by the quantile mapping,
so that the mismatches in concentrations have been removed or mitigated against, to ensure that such
differences cannot swamp the signal, as Fig. 6(c) suggests they would.

The distributions across all the identified forecast and observed objects can be analysed separately and
presented as box-and-whisker plots. Recall that the box encompasses the inter-quartile range (IQR, $25^{th}$
to $75^{th}$ percentile) and the notch and line through the box denotes the median or $50^{th}$ percentile. The
dashed line represents the mean, and the whiskers show $\pm 1.5$ times the IQR. For clarity, values outside
that range have been filtered out of the plots shown here.

Figure 9 shows a selection of AMM7v8 and L4 ocean colour product object attributes through the 2019
bloom season, such as individual object areas and intensity information (concentrations) as a function of
lead time (in days). Panel (a) shows the object areas (in model grid squares). AMM7v8 forecasts have a
broader size distribution and are generally bigger than the L4 objects. The mean (dashed line) is outside
the box denoting the IQR, suggesting that the area distributions are extremely skewed. The mean is
completely dominated by the large areas, even if they are few in number. The same is true for the L4
objects. Panels (b) to (d) try to provide some insights into the concentrations within objects. The lower
end of the concentrations (below the defined threshold) have been removed through the thresholding
process so that the distribution minimum here is defined as the object threshold. However, looking at
the $10^{th}$ percentile, $50^{th}$ percentile and $90^{th}$ percentile values of the within-object distributions (arguably
the part of the distribution of interest) can provide information on the concentration biases and the
general behaviour of the distribution, which is useful for aiding model development. It provides a
specific way of looking at the bias, which having been accounted for in terms of thresholding, is still
present within the objects. Figure 9(b) shows the range of 10th percentile concentration values within





the objects, i.e. above the threshold used to identify the object (in this case 2.5 mg.m$^{-3}$). [The concentration values within all the objects already *exceeding the threshold* are ranked and specific percentiles of the values *exceeding the threshold* can be extracted.] The 10th percentile *within-object* concentration values for the L4 ocean colour product are lower than those from AMM7v8, showing the bias, and their median and mean values are closer together. The 50$^{th}$ percentile of the *within-object* distributions shown in Fig. 9(c) displays similar behaviour but the difference between the L4 and AMM7v8 "median of medians" is even larger than in (b). The 90th percentile *within-object* concentrations in Fig. 9(d) show that the AMM7v8 and L4 distributions have, for the first time, similar median values, with the L4 ocean colour product having somewhat broader distributions and larger values, which is consistent with the apparent convergence in the distribution shown in Fig. 3. In addition, the observed means appear slightly larger than those forecast, reflecting the tendency for the L4 objects to reach higher concentration values, especially in coastal locations. To summarise there are three main messages from this figure:

- the AMM7v8 objects are too large, even when the bias is taken into account;
- the AMM7v8 concentrations are very biased, except in the tail, where they are more similar but not predicted often enough; and
- there is little to no change in the behaviour of the AMM7v8 forecast with lead time.


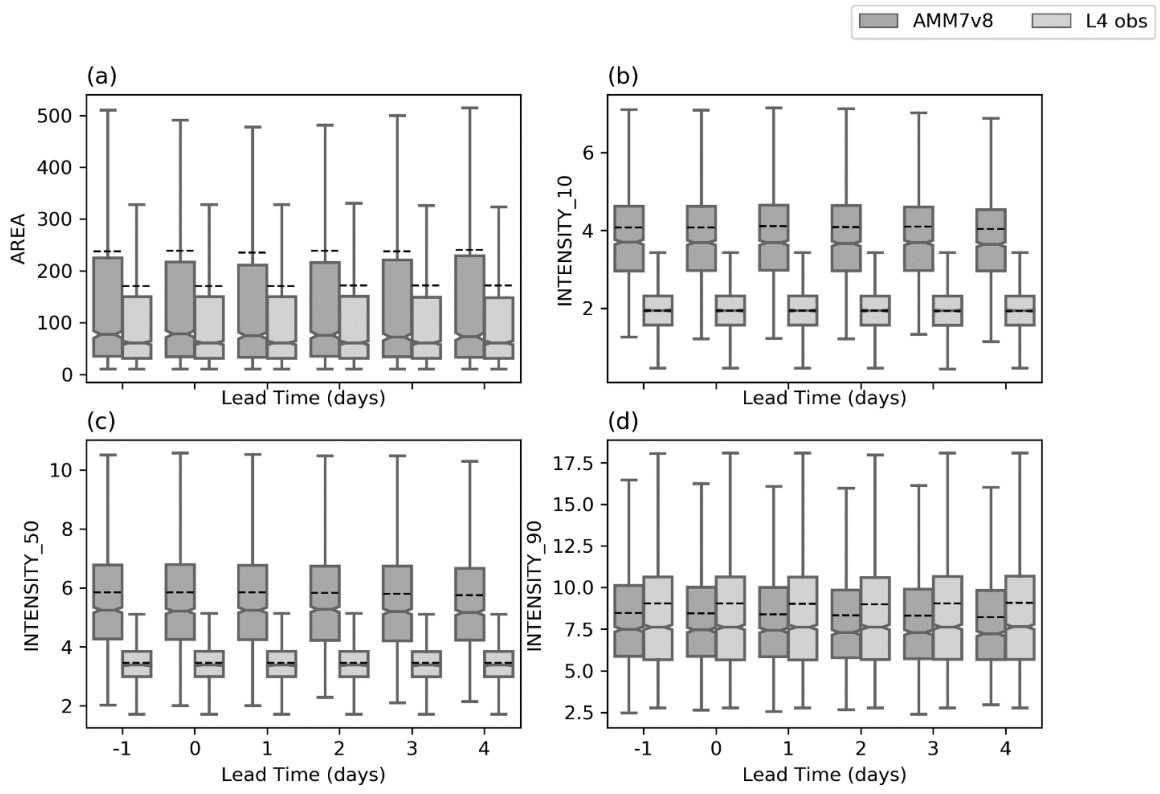

**Figure 9. Object attribute distributions for a) object area (in grid squares), b) 10th percentile of concentration values (above the threshold, in units of mg.m$^{-3}$), c) distribution of median concentrations (50th percentile) in units of mg.m$^{-3}$, and d) 90th percentile of concentration (above the threshold, in units of mg.m-3), for both the forecast objects (AMM7v8) and the observed objects (L4).**

The evolution of the number of objects identified through the 2019 bloom season is shown in Figure 10, illustrating how elements of the marginal and joint distribution information provided by MODE can be used together. Here both matched (joint) and unmatched (marginal) objects are shown. Both L4 ocean colour product and AMM7v11 analyses results are shown separately in (a) and (b). It is important to emphasise that even though the forecasts are the same in both (a) and (b), the different "truths" used could affect which AMM7v8 forecast objects are matched. There should be fewer unmatched objects than matched ones (ideally there would be no unmatched objects in either the forecast or the analysis). In Fig. 10 the number of objects in both sets of observations (AMM7v11 and L4) starts off small and increases as the bloom develops. In general, the number of matched forecast objects in Fig. 10(a) evolves in the same way as the number seen in Fig. 10(b). A spike in the number of matched objects seen in early April can be attributed to several coastal locations, which appear to be spatially well-


matched. In addition, a larger Chl-*a* bloom is seen in the Dover Straits region in the L4 ocean colour
product and although not exactly spatially collocated, the objects are matched. There are a consistently
large number of unmatched objects seen in the AMM7v11 analysis and L4 ocean colour product from
the end of May onwards. In the AMM7v11 analysis this appears to be due to an increase in small
objects identified, mainly to the west, north and east of the United Kingdom. The increase in unmatched
objects in the L4 ocean colour product is of a different origin, being due to an increase in localised
coastal blooms.

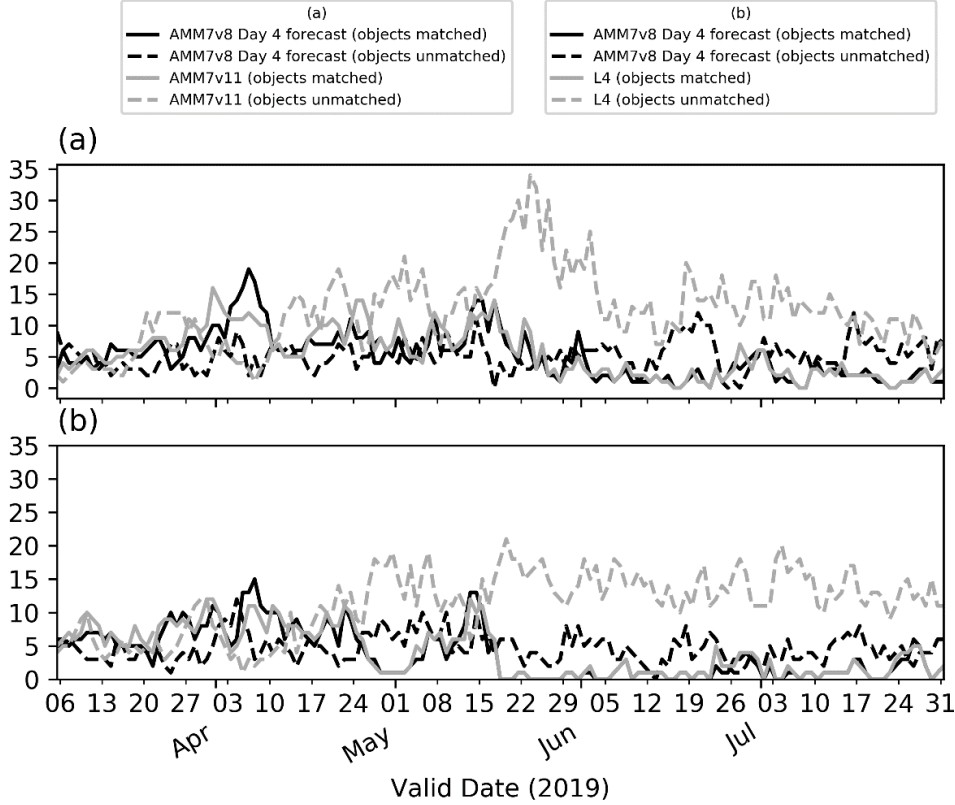


**Figure 10. Time series of the number of matched and unmatched objects from the MODE runs comparing (a) Day 4 AMM7v8**
**forecasts (black) with AMM7v11 analysis fields (grey) and (b) comparing Day 4 AMM7v8 forecasts (black) with L4 satellite**
**product observations (grey).**
The identified objects in each of the data sets: AMM7v8, AMM7v11 and L4 ocean colour product can
also be considered spatially, by counting the frequency with which a given grid square falls within an
identified object on any given day. These can be added up over the entire season to produce a spatial





composite object or "frequency-of-occurrence" plot. Figure 11 shows this spatial composite identified through the 2019 bloom season for each of the AMM7v8 Day 1 forecast objects (a), the L4 ocean colour product objects (b) and the AMM7v11 objects (c). All objects are identified using the 2.5 mg.m$^{-3}$ threshold. The AMM7v8 objects in (a) are clearly larger and cover more locations but each location with a lower frequency; there are more grid squares where there is an object identified between 0–20% of the time than for the L4 observed objects, as seen in Fig. 11(b). Noticeably, there is a patch in the central North Sea where the AMM7v8 forecasts identify objects some of the time, but the L4 ocean colour product does not have objects there at all. The AMM7v11 analysis, shown in Fig. 11(c) has objects there some of the time, but looks more like the L4 composite; this could indicate the model tends to generate high Chl-*a* concentrations in this area, but the data assimilation is able to constrain it.

However, there are areas, for example in the South West Approaches, where there appears to be a good level of consistency between the forecast and observed object frequencies. AMM7v11 has elevated Chl-*a* values along the northern and western edges of the domain, for a low proportion of the time, which are not seen in the L4 product, and are also different to AMM7v8. This is likely due to changes in how the nutrient and phytoplankton boundary conditions have been specified between AMM7v8 and AMM7v11, due to Chl-*a* being too low near the boundaries in AMM7v8. The advantage of assimilation of the satellite observations within the AMM7v11 analysis can be seen around the coast; proportions in Figure 11(c) have moved to similar levels as seen in the L4 plot (b) in coastal locations.

Figure 11 also shows that even when the differences in concentrations (bias) are accounted for in the thresholding, the extent or size of the AMM7v8 forecast objects (which represent the Chl-*a* blooms) is still overestimated compared to the L4 ocean colour product with the AMM7v11 analysis sitting somewhere in between these two solutions.




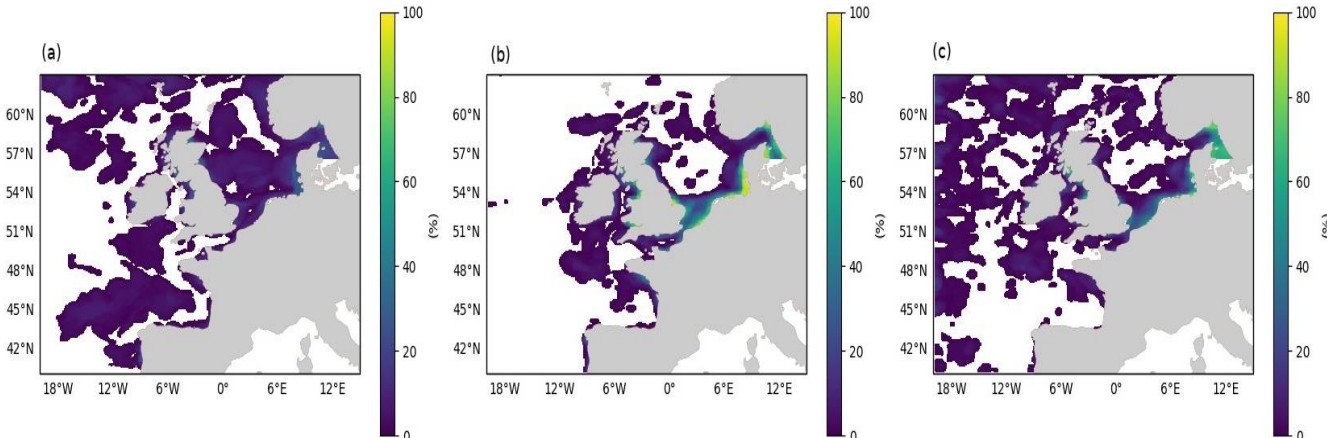

**Figure 11 Object composites (the proportion of time for which an object was present at the grid box throughout the 2019 bloom season) for a) the AMM7v8 day 1 forecast objects, b) the L4 ocean colour product objects and c) the AMM7v11 analysis objects. For (a) the thresholds varied to but were anchored to "truth" threshold of 2.5 mg.m$^{-3}$, which was used for (b) and (c).**

Thus far all the attributes have been based on only the forecast or only the observed objects. Figure 12 gives an example of a paired object attribute using box-and-whisker plots, which are produced by comparing the AMM7v8 day 0 forecast to L4 and AMM7v11 (labelled AMM7v8 vs AMM7v11, and AMM7v8 vs L4) and a third option of comparing the two truth sources (labelled AMM7v11 vs L4). Figure 12 shows the intersection-over-area diagnostic, which essentially gives a measure of how much the paired forecast-observed objects overlap in space. If the objects do not intersect, this metric is 0. The IQR is ~0.45 with 50% of paired objects having an intersection-over-area of 0.6 or greater (it is easy for smaller L4 ocean colour product areas to be completely enveloped by the model analyses, even with the concentration bias accounted for). However, the whisker spans the entire range of values (between 0 and 1) which shows that there are instances where this metric is 0. It clearly shows that the AMM7v11 analysis is closest to the L4 ocean colour product, with all pairs overlapping in some way. Finally, the AMM7v11 vs L4 shows the most compact distribution of values. There is quite a difference between the median (notch) and the mean (dashed line) for this metric, suggesting the distribution is skewed with the mean affected more by many small values.





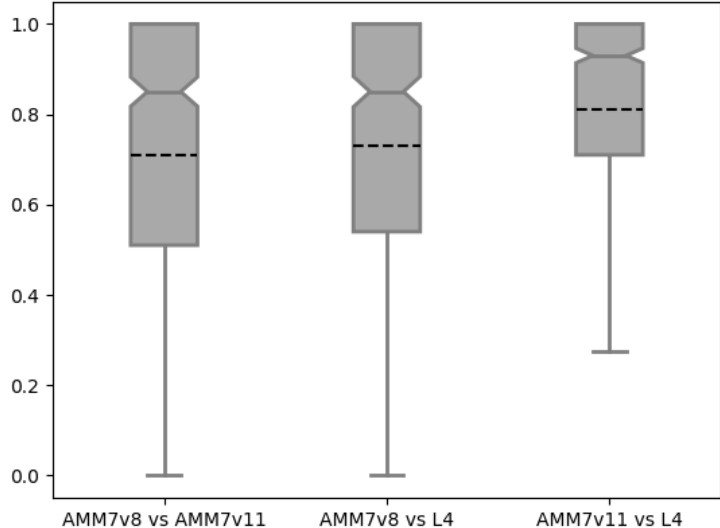

635

**Figure 12. Ratio of the intersection area over the largest of the forecast or observed object area for Day 0 Chl-*a* concentrations exceeding 2.5 mg.m$^{-3}$ (for L4 ocean colour product and AMM7v11) and a smoothing radius of 5 grid squares.**

638

## 4.4 Location errors

640

The focus shifts to MTD output in subsequent sections. Having information in space *and* time enables one to ask, and hopefully answer, many questions related to how the bloom season was initially detected and subsequently forecast. What is particularly helpful is that elements such as location and timing errors can be treated separately to answer: *"did the model predict the bloom to start in the observed location?"* or *"did the model predict the onset at the right time?"* and "*did the model predict the peak and duration of the bloom correctly?"*. We address location errors first.

647

Recall that objects are now identified in space and time. Recall also that a manual quantile mapping was used here as a more automated method was not available for MTD. As previously described, all MTD results are based on a 2.5 mg.m$^{-3}$ threshold applied to the L4 ocean colour products or AMM7v11 analyses and a 6 mg.m$^{-3}$ threshold to the AMM7v8 forecasts. First, the location error of the blooms is examined using the *time* centroid for a space-time object. This time centroid is derived from a time



series of spatial (two-dimensional) centroids which are extracted for each time slice, and which represents one of the inputs to identifying the 3-D space-time objects. The time centroids for all identified MTD objects during the 2019 bloom are shown in Fig. 13. The filled circles represent the observed time centroid (large represent AMM7v11 and small reflect L4). All other coloured symbols indicate the AMM7v8 forecast time centroids. The colours represent the relative position within the season, with blue (cool) colours early in the season (March onwards), and the reds and pinks (warm colours) towards the end of the season (July). The forecast time centroids for the different lead times are essentially on top of each other showing there is no variation with lead time in the centroid position. The impact of using the AMM7v11 analysis and L4 product is evident in the observed centroids, with the AMM7v11 analysis in (a) producing many more objects in deeper waters to the north and west of the domain.

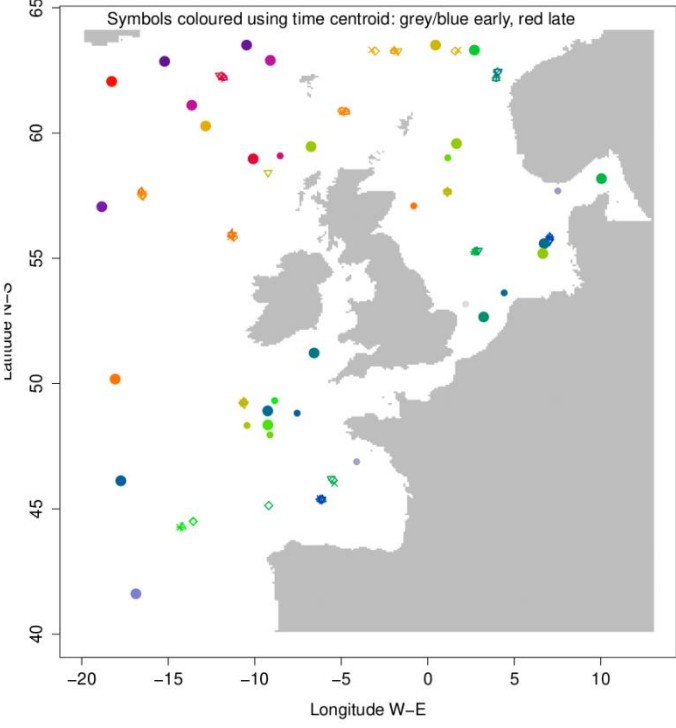

**Figure 13. Time centroids for the simple objects identified from AMM7v8 forecast objects (various symbols), AMM7v11 (large filled circles) and L4 ocean colour product (small filled circles). Colours reflect approximate position in the season and also highlight the north- and westward progress of bloom over time. Also refer to Fig 16 for colour cross reference and Fig 14 for the forecast lead time symbols.**



## 4.5 Onset and evolution

A time series of all identified object areas (the 2-D individual time slices making up the 3-D space-time MTD objects) is plotted in Fig. 14. Being able to distinguish between the different objects is not relevant at this stage. In Fig. 14(a), all the L4 ocean colour product objects' areas are in black and all the AMM7v11 objects' areas in grey. There can be (and are at times) more than one space-time object on any given day. In (a) both sets of objects were identified using thresholds of 2.5 mg.m$^{-3}$. The first identifiable Chl-$a$ bloom object in the AMM7v11 analysis was identified on 29 March 2019 whereas in the L4 ocean colour product this was on 3 March, 26 days earlier. In (b) the black dots representing the L4 ocean colour product are the same as in (a). The different AMM7v8 forecast lead times are indicated by the different coloured symbols. On each day there are 5 coloured symbols for each object that exists on that day and for AMM7v8 a threshold of 6 mg.m$^{-3}$ was used, which as described earlier, is based on the CDF for the whole season. The AMM7v8 forecasts only picked up the first event of the season on 18 April 2019, which is another 20 days later. Subsequent events (represented by the objects) are somewhat better aligned in time but the mid-May peak is primarily associated with what could be a classified as a false alarm where AMM7v8 produces a substantial bloom to the SW of the UK which was not observed.

The fact that all the forecast lead time symbols are very closely collocated on each day confirms that there is very little difference in the forecast areas as a function of lead time. The L4 ocean colour product also suggests that the bloom ends 30 June whereas both the AMM7v11 analyses and AMM7v8 forecasts persist the space-time objects to 23 July and 14 July respectively. Taking the start of the earliest space-time object as the onset of the bloom season and the end of the last object as the end, the 2019 season is 119 days long, based on the L4 product, 117 days in the AMM7v11 analysis and 87 days in AMM7v8. Therefore, the length of the season is comparable in the AMM7v11 analysis, albeit with a large offset. The AMM7v8 model produces a short and intense season which starts ~1.5 months (46 days) too late and persists 2 weeks beyond the observed end.





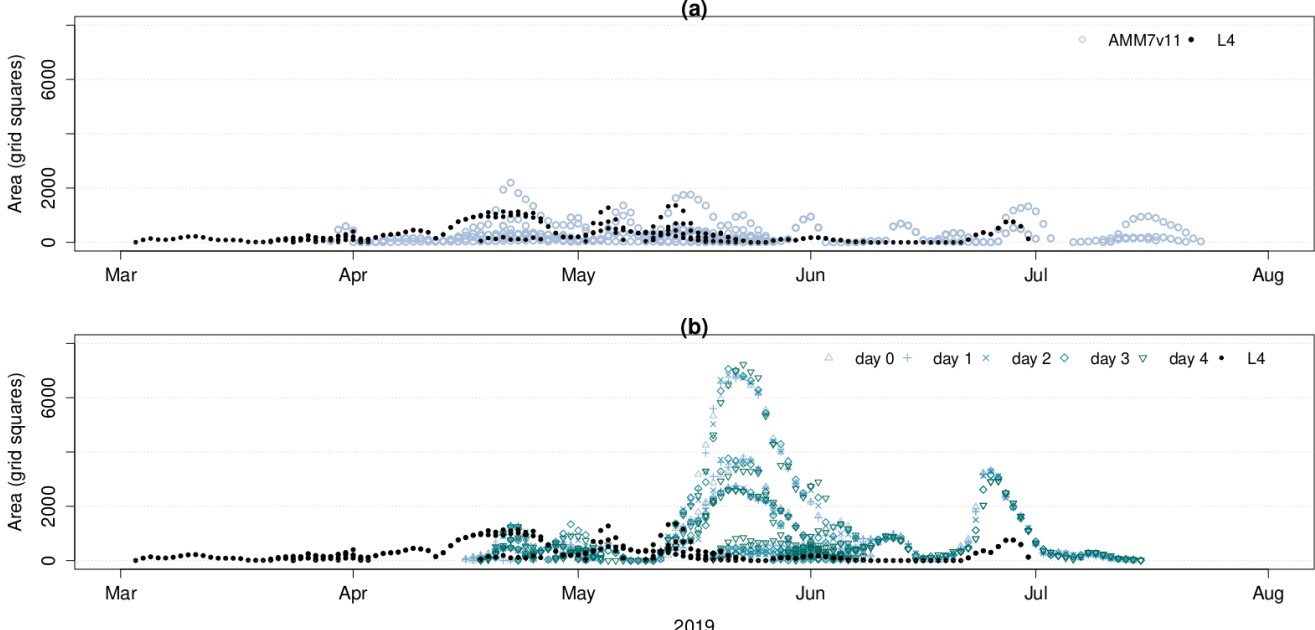

**Figure 14. Time series of all identified single simple MTD object areas. (a) AMM7v11 analysis and the L4 ocean colour product object areas, further confirming that for the most part the AMM7v11 analysis behaves more like the L4 product in both concentration and spatial extent of objects. (b) Comparing the AMM7v8 forecasts to the L4 ocean colour product objects (which are the same as in (a)), showing the mismatch in timings, in terms of onset of the bloom season as well as the mismatch in bloom extent.**

The temporal evolution of the Chl-*a* blooms during the 2019 season can also be viewed spatially as shown in Fig. 15. The space-time objects are shaded by object number with numbers increasing from the start of the season, showing how the bloom migrates north and westwards as the season unfolds. In (a) the L4 ocean colour product objects are shown, in (b) AMM7v11 and (c) shows the day 4 AMM7v8 forecast objects. The L4 product in (a) has the fewest and smallest identified objects. The AMM7v8 in (c) produces fewer but much larger objects (blooms). The AMM7v11 analysis in (b) sits somewhere in the middle in terms of number of objects and extent. Both AMM7v8 and AMM7v11 provide poor information in the NW of the domain. This area is heavily influenced by the biogeochemical (climatological) boundary conditions, and fortunately not that relevant to users, who are primarily interested in the on-shelf region.





(a) (b)

(c)

**Figure 15. Temporal evolution of identified Chl-*a* single simple MTD objects, based on the daily sequence of either the L4 ocean colour product (a), the AMM7v11 analysis (b) and the AMM7v8 day 4 forecast in (c). Colours correspond to the object numbers assigned by MTD but also act as a proxy of time.**





The progression of the bloom can also be viewed purely from a temporal perspective, as shown in
Figure 16 (a) and (b), providing a clearer view of the onset and demise of each object (bloom episode),
compared to that provided in Figs 12 or 13. The x-axis represents elapsed time. Vertical lines on any
given date indicate the temporal location of a time centroid and the initial identification and end of a
given object/event are indicated by the start and end of the vertical lines. Solid lines represent the
observed events (in either the L4 ocean colour product or AMM7v11) whereas dashed lines are the
forecast events, which are the same in (a) and (b). From this the difference in the onset of the 2019
season is very clear. Most forecast objects are of relatively short duration, but overall, most groups of
forecast objects have some temporal association with an observed object around the same time (though
this does not mean they are close in space).

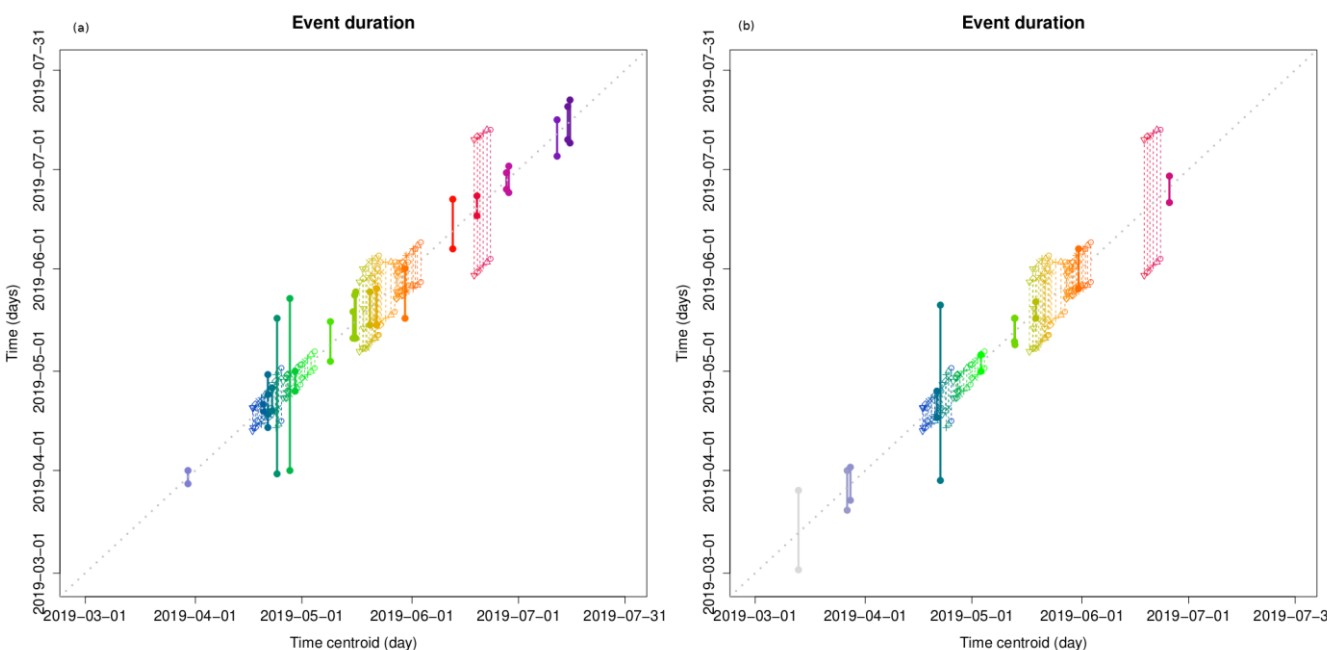


**Figure 16. Duration of single simple time objects and their location relative to the start of the Chl-*a* bloom season for AMM7v8**
**(dashed line) and (a) AMM7v11 and (b) L4 ocean colour product as observations.**





## 4.5 Examining the MTD paired object attributes

From Fig. 15 it is clear that there are relatively few space-time objects within the 2019 bloom season and MTD only identifies 13 matched object pairs based on using simple single objects. This makes drawing any robust statistical conclusions somewhat difficult. Nevertheless, a selection of paired object attributes is presented in Fig. 17 for AMM7v8 day 4 forecasts compared to the AMM7v11 analysis. The different shadings indicate groups of attributes which are similar to each other, i.e. relating to distance, time or volume. From the figure we can conclude the following:

- The spatial centroid (centre of mass) differences can be extensive, but the majority are within 0 to 50 grid squares apart (i.e. up to ~350 km).

- The majority of paired objects have time centroid differences +/- 20 days of the observed, with a preference for the forecasts being later (difference being defined as forecast time minus observed time). This is better illustrated by the distribution of start and end times. In terms of the event duration forecast blooms are generally too short.

- Generally, the orientation of objects is within 40 degrees.

- There is a fairly even split in terms of the spatial speed of propagation of the bloom, though it is hard to infer whether there is a specific fast or slow bias.

- Considering the volumes of the space-time objects, the majority of objects have volume ratios of less than 5 (forecast-to-observed ratio), i.e. AMM7v8 objects tend to be *much* larger, but despite this only one bloom episode provided a large overlap in space and time. In other words, despite the size of the forecast objects, the paired objects are sufficiently far apart (in space and/or time) they still do not overlap.


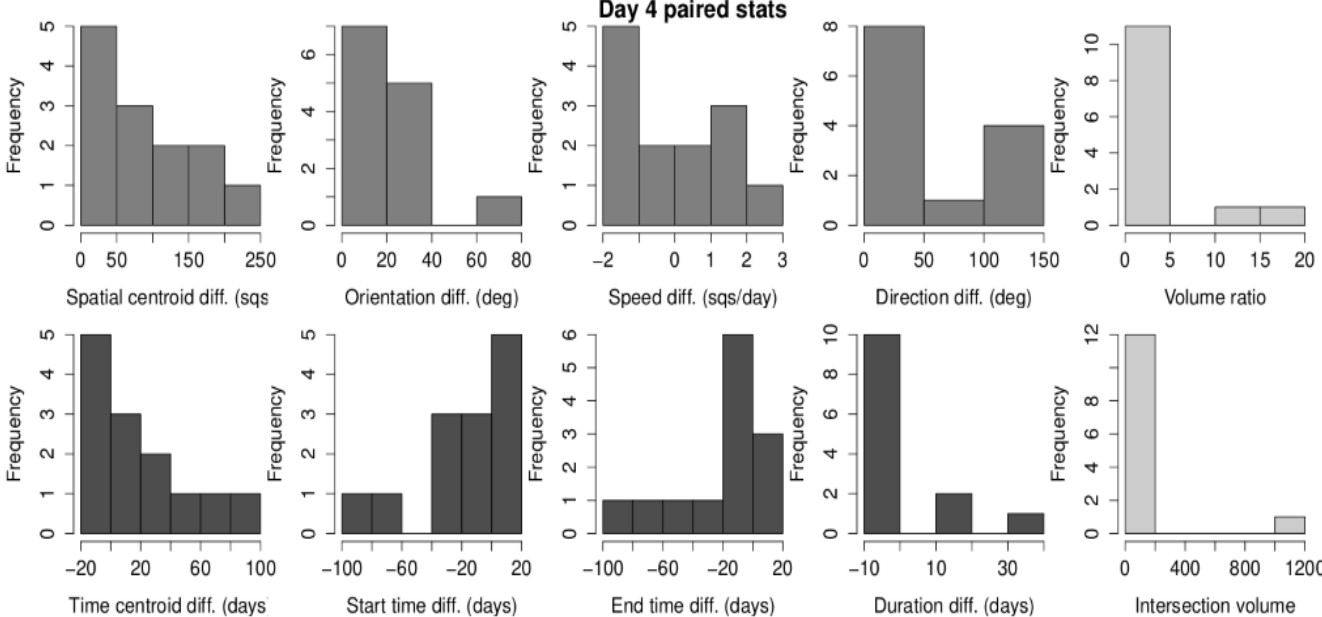

**Figure 17**. **Summary of MTD simple pair forecast-analysis object attributes based on using the AMM7v11 analysis as the verifying analysis. Here the day 4 forecast results are shown but the results are very similar for all lead times.**

## 5. Conclusions

MODE and MTD were used with the operational AMM7v8 European North West Shelf Chl-*a* concentration forecasts to evaluate whether the objects (blooms) produced were similar in structure, location and timing to those produced by the L4 ocean colour product. The pre-operational AMM7v11 model analysis, which includes assimilation of Chl-*a* observations (referenced here as AMM7v11) was also assessed.

There is a significant concentration bias in the forecasts compared to the satellite ocean colour product. This needs to be mitigated against before using a threshold-based methodology such as MODE or MTD, which aims to understand the spatial properties of the forecasts (i.e. the spatial extent is affected). A quantile mapping approach was used to mitigate against this concentration bias to ensure that the frequency of occurrence of specific concentrations remained the same, either precisely (for MODE) or



approximately (for MTD, where the seasonal CDF was used to estimate approximately equivalent
concentrations). Blooms were said to occur when the observed concentration threshold exceeded 2.5
mg.m$^{-3}$. Forecast thresholds for MODE were then relative to this value and varied from day-to-day. For
MTD the seasonal equivalent threshold for the AMM7v8 forecasts was 6 mg.m$^{-3}$.

With the impact of any concentration bias being mitigated against, MODE results suggest that the
forecast blooms are too large; this spatial extent bias is in addition to the concentration bias noted
above. As well as forecast objects generally being too large, AMM7v8 produces more objects (in
number) than seen in the L4 ocean colour product, yet many of the coastal objects seen in the L4
product cannot be resolved by the model due to the coarseness of the coastline in the 7 km model. This
situation would improve should the model resolution increase from 7 km to 1.5 km.

The lack of variation of results with increasing lead time is important to note. For all forecast lead times
out to day 4 there was no significant change in results for any of the thresholds analysed. This could be
an indication of the processes involved acting on timescales longer than this, or it could be an indication
of a deficiency within the model. In addition, predicting the onset of a Chl-*a* bloom seems problematic
for the model as it currently stands (AMM7v8), with the forecast being 46 days later than observed. The
AMM7v11 analysis reduced this to 26 days, so it would be reasonable to expect that when forecasts are
initialised from this analysis in the future, that the lag in the onset will be reduced significantly. The
AMM7v8 forecasts reflect a model climate which wants to produce a shorter and more intense season
than what is observed. The model also struggles with predicting the end of the season, being around 2-3
weeks later than observed, suggesting that AMM7v8 blooms persist too long compared to those in the
L4 ocean colour product.

Once AMM7v8 has picked up the start of the season, subsequent events are handled somewhat better.
Beyond the timing issues, the model does generally produce Chl-*a* blooms in roughly the right locations
but not necessarily at the right time, though the overlap between blooms can still be limited, despite the
apparent size advantage of the AMM7v8 bloom objects.





Constraining the Chl-*a* using assimilation of the satellite observations appears to benefit the model in
terms of less unmatched bloom regions; an improvement in the forecasts generated from this analysis is
expected and will be the subject of future work.

## 803 6. Code availability

Model Evaluation Tools (MET) was initially developed at the National Center for Atmospheric
Research (NCAR) through grants from the National Science Foundation (NSF), the National Oceanic
and Atmospheric Administration (NOAA), the United States Air Force (USAF) and the United States
Department of Energy (DOE). The tool is now open source and available for download on github:
https://github.com/dtcenter/MET.

## 810 7. Data availability

Data used in this paper was downloaded from the Copernicus Marine and Environment Monitoring
Service (CMEMS). The datasets used were:
• https://resources.marine.copernicus.eu/?option=com_csw&task=results?option=com_csw&view=de
tails&product_id=OCEANCOLOUR_ATL_CHL_L4_NRT_OBSERVATIONS_009_037 (last
access: August 2019),
• https://resources.marine.copernicus.eu/?option=com_csw&view=details&product_id=NORTHWES
TSHELF_ANALYSIS_FORECAST_BIO_004_002_b (last access: August 2019)

The AMM7v11 analyses are not operational and not yet available from the CMEMS server.

## 820 8. Author contribution

All authors contributed to the introduction, data and methods, and conclusions. MM, RN, JM and CP
contributed to the scientific evaluation and analysis of the results. MM and RN designed and ran the





model assessments. CP supported the assessments through the provision and reformatting of the data
used. DF provided detail on the model configurations used.
**9. Competing interests**
The authors declare that they have no conflict of interest.

**10. Acknowledgements**
This study has been conducted using E.U. Copernicus Marine Service Information.

This work has been carried out as part of the Copernicus Marine Environment Monitoring Service
(CMEMS) HiVE project. CMEMS is implemented by Mercator Ocean International in the framework
of a delegation agreement with the European Union.

We would like to thank the National Center for Atmospheric Research (NCAR) Developmental Testbed
Center (DTC) for the help received via their met_help facility in getting MET to work with ocean data.

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
