# Peer review of "Using feature-based verification methods to explore the spatial and"

_Ocean Science, 2020_

## Referee Comment (RC1) · Anonymous Referee #1 · 11 Dec 2020

The authors apply a feature-based comparison technique to sea surface chlorophyll snapshots in order to compare a model forecast to satellite observations and data-assimilative model output. While potentially useful, the particular forecast that is used in the study performs so poorly that the sophisticated comparison metrics cannot be assessed properly.

**general comments**

I have two major comments and the first likely affect the second. While the manuscript

starts out well written, sections 3 and especially 4 are very difficult to follow. Some statements seem to contradict earlier ones and I found myself second-guessing what I had just read in light of the next paragraph.

A good example of this occurs in section 4.3: "All objects are identified using the 2.5 mg mˆ-3 threshold." (l 594). Just as I was about to ask why a constant threshold was used again (when it does not work in earlier examples), the caption of Fig 11 states that "For (a) the thresholds varied to but were anchored to "truth" threshold of 2.5 mg mˆ-3". So it appears that variable thresholds were used but this is far from immediately clear. Yet this information is important to understand the results.

Generally, terms need to be introduced better and used more consistently: In section 4.1, an "observed threshold" is introduced, followed by a "forecast threshold", and a "seasonal threshold", but these terms are not used consistently. Similarly, "forecast" and "analysis" need to be introduced better: Initially the manuscript states that "In order to assess the European NWS Chl-a concentration forecast (AMM7v8), a satellite based gridded ocean colour product (L4) product and model assimilative analysis (AMM7v11) are considered as gridded "truth" sources." But then, without introduction, an AMM7v8 analysis appears and from then on "AMM7v8" may refer to the forecast or the analysis. To add to the confusion, the plural term "AMM7v8 forecasts" or just "forecasts" is often used, which (I think) is referring to the sequential nature of assimilative forecasting but is not helpful here to the reader.

To improve the description of MODE/MTD I would also recommend to include some early examples of what objects typically look like in the context of sea surface chlorophyll. Currently, the MODE algorithm is introduced, object attributes are described, MODE tuning is examined, difficulties applying MODE are identified, and an object's minimum size is discussed (pages 8-10) without ever mentioning what a typical object would be. It would be very useful for the reader to include an example image, a bit more zoomed in than those in Fig 2.

As a reader interested in models and model assessment, my main interest in the manuscript was to see how the comparison technique works, and performs in a typical scenario. Unfortunately, it appears to me that the the model that was selected here underperforms so much in terms of recreating surface chlorophyll (both in magnitude and timing) that any far less sophisticated comparison would show it. Worse yet, the large discrepancy between AMM7v8 and the data appears to hinder a high level comparison of the two. So in the end, as a reader, I know that AMM7v8 does not recreate surface chlorophyll well but I am not sure if MODE or MLT are well suited for assessing surface chlorophyll output.

Here, I would suggest the following: rather than focusing on the comparison of AMM7v8 to L4 and AMM7v8 to AMM7v11, the manuscript could focus much more on the comparison of AMM7v11 to L4. A drawback here would be that L4 was already used to inform the AMM7v11 analysis but AMM7v11 performs much better thus permitting a much more interesting high level comparison. Perhaps a AMM7v11-based forecast could be employed to assess the capabilities of MODE and MTD.

**specific comments**

l 21: "whilst several forecast blooms did not materialise in the observations": This sounds like the data is to blame for not showing the bloom, maybe rephrase to something like: "while the model forecasts also showed blooms that do not appear in the observations".

l 22: "Whilst the model...": Most of this has been said in the previous sentence, would be more useful to move it one sentence up as an intro.

l 34: "double penalty effect": For readers not familiar with this term, it would be beneficial to describe this term a little better. Currently it sounds like not in the right place would be a single penalty and additionally not at the right time would be the double penalty.

[Figure]

l 114: "the models at 7 km resolution cannot resolve the coasts": It is not entirely clear what exactly is meant here: are you referring to the coastal chlorophyll a dynamics?

l 126: "is" -> "are"

l 133: It would be good to introduce the "Atlantic Margin Model" the first time "AMM" is used in l 117 or l 98.

l 135: I have a little trouble understanding this: is "Day 4 for the period of 1 March-31 July 2019" simply March 4th or are multiple 4-day forecasts created?

l 136: Mention that both the analysis and the forecast are used in this study. The previous sentence is confusing to the reader, it appears to state that the forecast is hereafter referred to as AMM7v8, when later on AMM7v8 forecast and analysis are used.

Fig 1: I would prefer to have the log scale included in the colorbar ticks ("10ˆ{-3}" instead of "-3") rather than written out in the title. This would also eliminate potential confusion, as the values in (d) are not exponents, although the title may indicate that.

l 150: Use (a) and (b) instead of "left" and "right". Same for the next sentence.

l 158: "can be comparable": That is a very imprecise statement, as a reader I am not sure what this is telling me. Are the observation errors of the same magnitude as the model bias?

l 166: The instrument to measure ocean colour is satellite-borne, the ocean colour is not. Maybe use "remotely-sensed" or "satellite-derived"? Also, I would not refer to colours as concentrations.

l 167: Out of curiosity, why is a different satellite product used for the comparison and is there a significant difference between the two satellite products?

l 172: Is this referring to the coupling between physical and biological model components?

l 173: Is the AMM7v11 data assimilation also based on 3Dvar?

l 181: The "sequence of forecasts" is a bit confusing here. The next sentence refers to "a forecast". I know that data assimilation can create a sequence of forecasts but here I think it would be much easier for the reader to stick with "a forecast" throughout the description of MODE.

l 184: The mention of a model-based analysis is not helpful to the reader here, and it also not used again in this paragraph. I would suggest to rephrase to something like: "in this context, one is typically a model forecast, the other an observed field, i.e. observations regridded to match the locations of the model grid." Later on it can be mentioned that it can also be used to compare two model fields.

l 189: "observed objects" -> "observed field"

l 198: "and is based on a disk": Is the convolution kernel really a (flat) disk? The "based on" is confusing here.

l 201: Maybe use "objects" again, as above, instead of "areas".

l 204: Are the observed fields not smoothed? In step 2 it sounds like both fields are smoothed.

l 212: This is unclear: the first sentence in (6) "which together are expressed as the so-called "interest" score" makes it seem like one interest score is computed summarizing the fit of all objects. In this sentence, "interest scores are computed for all" objects. Please rephrase.

l 252: "minimum volume of 1000 grid squares": This should be "area" or is this applied below the ocean surface? Why is it 1000 grid squares here and 10 above? Reading it a few more times, it seems like this describes a "volume" in space-time, which needs to be made more explicit. "1000 grid squares" is not a volume, is this 10 grid squares times 100 time steps?

l 255: I am guessing here that paired objects are those for which an equivalent was found in the other field while all others are called single. And clusters are two or more objects in one field, classified as belonging together. Please add a bit of description here, to define these 4 terms.

l 262: Using the log-transformation seems like a sensible choice for chlorophyll. It would be good to know if the transformation was applied before smoothing the fields.

l 266: "mg." -> "mg"

l 267: The units here would be "log_10(mg mˆ-3)". In my opinion, it would be clearest to just provide the values in linear space, e.g.: "For this study, a range of thresholds were applied to the log10-transformed chl-a fields, corresponding to chl-a concentrations between 1.62 and 25 mg mˆ-3."

l 268: Why use 25 mg mˆ-3 here when the values of interest are in the range 3 - 5 (previous sentence)?

l 274: "This radius" -> "The 5 grid point radius"

l 276: Why so much discussion of different thresholds above when only a single one will be used here? Or is "here" only referring to the sensitivity analysis? Please be more specific.

l 304: What is "this", maybe use "The effect of bias"

l 310: "would yield no useful information": One could argue that having no matched pairs contains the useful information that the model solution cannot be very good.

l 315: Why is the L4 product referred to as an analysis here?

l 324: "the two that clearly differ more dramatically from Fig 2": What is meant here is probably "the two fields that show the largest discrepancies in Fig 2".

l 336: What about using different thresholds for the different fields?

l 339: "forecasts": Which forecast, only analysis solutions were considered thus far.

Fig 3: Please include the AMM7v8 distribution after the quantile mapping. And why not include the AMM7v11 distribution as well?

l 356: "the observed threshold": Do you mean the threshold applied to the observations?

l 358: After a lot of reading the next paragraphs over and over to figure out what the thresholds are, this is the position where I lost track. Here, it needs to be stated that the "value that has the equivalent rank in the forecast distribution" is the "forecast threshold". They are currently not linked and initially I thought that the forecast threshold was the same as the (slightly misnamed) observed threshold.

l 362: Based on this description it is not clear what this threshold is. It is also called "seasonal" here when Fig. 4 shows daily variations in a threshold. Are these the same thresholds, why is it called seasonal? If this is the threshold used to identify objects, please mention this explicitly.

l 365: Is this procedure applied to both v8 and v11?

l 374: "the latter": is this the L4 product? "the latter" is not really referencing anything at this point, please just use the product name.

l 375: Please explain better what this threshold means. It is impossible to understand this paragraph without knowing what the threshold signifies.

Fig. 4: It would be good to include a grid and reduce white space, the threshold never exceeds 6. I would further suggest to merge this figure with Fig 5.

l 390: "Error! Reference source not found." Something appeared to have gone wrong with the automated submission system.

l 391: "using the built-in functionality in MODE as for Fig 4.": This is unclear, please rephrase.

l 394: What is the forecast lead time?

l 405: "the forecasts are very active": What exactly does this mean, phytoplankton blooms?

l 406: "The latter object is not identified in the L4 ocean colour product." It is not clear what "object" this is referring to. I do not think this paragraph is very helpful to the reader at this point. So far the reader has only seen an example of what object identification should not look like (Fig. 2) and now here is a lengthy explanation of chlorophyll-a blooms leading to peaks in a threshold without showing an example of that these variable thresholds improve object identification.

l 434: "this work was done without accounting for the concentration differences": Does this imply that the variable threshold from the previous section was not used? But it seemed to be very import for successful identification.

l 444: Are "quilt plots" the same as "quilt "difference" plots"? I would suggest to stick with one term.

l 444: Here are two nearly identical sentences two (short) paragraphs apart: "Figure 6 provides a selection of quilt plots derived from using the L4 ocean colour products and AMM7v8 analyses during July 2018, using one of the merging options which was tested." (l 435) and "In Figure 6 some quilt "difference" plots are shown to focus on the individual characteristics of the AMM7v8 analysis and the L4 ocean colour product based on a set of initial data that was available for July 2018."

l 451: How useful is this analysis if it is already clear from the previous section that a comparison at the same threshold is not sensible? The 2.5 mg mˆ-3 threshold can be established without considering the model output.

l 475: Again, how valid are these results if different thresholds would be used for model and data? Would the conclusions about the smoothing radius hold?

Fig 6: The font is much too small.

l 489: Is this analysis done at the same threshold for data and model?

l 491: Is this "total area" of identified objects or ocean grid cells?

l 534: The lead time has still not been properly explained.

l 545: So these percentiles are characterizing the chl-a distributions among different areas. How can they contain values below the 2.5 threshold applied to the observations? The "(in this case 2.5 mg m^-3)" makes it sound like the same threshold was applied to the two products but this seems to contradict earlier sentences.

Assuming now that the threshold are different: From Fig. 3, we already know the distribution of chl values, what new information does Fig 9 give the reader? We know of the bias and have a rough idea of the distribution, and we also know that a higher threshold will likely be used for AMM7v8, which appears to be the main contributor to the differences between the distributions in Fig 9.

l 574: It would be nice for comparison to see good results for comparison. How would Fig 10 look for a AMM7v11 to L4 comparison?

l 623: "AMM7v8 day 0 forecast to L4 and AMM7v11 (labeled AMM7v8 vs AMM7v11, and AMM7v8 vs L4)": The labels are either switched or cleverly selected to confuse the reader.

l 653: Is the spatial centroid the average location of the center of a series of identified objects? This needs to be explained.

---

## Referee Comment (RC2) · Anonymous Referee #2 · 20 Dec 2020

Review of "Using feature-based verification methods to explore the spatial and temporal characteristics of forecasts of the 2019 Chlorophyll-a bloom season over the European North-West Shelf" by Mittermaier et al.

The paper investigates possibility to exploit a feature-based verification method (initially introduced and used for numerical weather prediction model evaluations) to assess forecast and analysis of chlorophyll-a concentration (Chl-a) in the European North-West Shelf (NWS) which is provided on regular basis by the UK Met Office. With the proposed verification method, the authors evaluated the forecast provided by the Met Office Atlantic Margin Model (AMM) with and without data assimilation against observed Chl-a satellite data (a product of the Copernicus Marine Environmental Monitoring Service, CMEMS). Given this subject, the paper would definitely fit the scope of the journal, and could be published after a revision. A further revision is, however, required. Below I provide several comments and suggestions the authors might want to consider for revising the manuscript.

**General comment**

To illustrate advantages of the presented verification methods (newly applied for Chl-a forecast evaluation), it would be nice to compare the proposed method with other metrics like, for instance, bias (classical), MAE(MAD), bloom phenological indices (Siegel et al. 2002, Soppa et al. 2016), or any methods previously used for AMM NWS Chl-a valuation (mentioned in lines 83 - 85) with respect to complication/simplicity, possible diagnostic (meaning) and conclusions drawn by the analysis.

**Specific comments**

Lines 15 – 16: I like that the discussed verification method allows not only identify bias but correct (mitigate its impact when carried other further analysis). How would it compare with regular bias correction?

Lines 526 – 530: Reads as this paragraph should not be at this place. Which figure is discussed? Figure 12?

Lines 553 – 557: How would it compare with analysis of the frequency distribution (histogram)?

Part 4.5 "Onset and evolution", I would suggest change the title to "Bloom onset and evolution". I am also curious how would the material and results presented in this subsection compare with simple bloom phenology analysis based a threshold method (Siegel et al. 2002, Racualt et al. 2012, Brody et al. 2013, Soppa et al. 2016).

> Brody, S.R.; Lozier, M.S.; Dunne, J.P. A comparison of methods to determine phytoplankton bloom initiation. J. Geophys. Res.: Oceans 2013, 118, 2345–2357.
>
> Racault, M.F.; Le Quéré, C.; Buitenhuis, E.; Sathyendranath, S.; Platt, T. Phytoplankton phenology in the global ocean. Ecol. Indic. 2012, 14, 152–163
>
> Siegel, D.; Doney, S.; Yoder, J. The North Atlantic spring phytoplankton bloom and Sverdrup's critical depth hypothesis. Science 2002, 296, 730–733. Doi: 10.1126/science.1069174

Soppa, M. A., Völker, C. and Bracher, A. (2016): Diatom Phenology in the Southern Ocean: Mean Patterns, Trends and the Role of Climate Oscillations , Remote Sensing, 8 (420), pp. 1-17 . doi: 10.3390/rs8050420

Lines 772 – 774: "Blooms were said to occur when the observed concentration threshold exceeded 2.5 mg.m$^{-3}$. Forecast thresholds for MODE were then relative to this value and varied from day-to-day. For MTD the seasonal equivalent threshold for the AMM7v8 forecasts was 6 mg.m$^{-3}$". What can be in general concluded from this about AMM7v8 performance?

**Typos**

Line 99: delete on of two words "product".

Line 187: "*merged*", "*matched*" – is italic font urgently required. Or it is a format error?

Line 197: "*further*" - a format error?

Line 206: "*not*" - a format error?

Line 257: *"single simple"* - a format error?

Lines 217, 218, 220, 232, 233, 237, 238, 242: please check/confirm if italic font used for a number of words is required (?since refer to specific options? right?).

Lines 546 – 548: again, a format error – used square brackets, *"exceeding the threshold"*, *"within-object"*

Lines 550 – 552: *"within-object"* - a format error?

Line 641: *"and"* - a format error?

Line 652: *"time"* - a format error?

**Figure quality**

Figure 2: please improve quality of the subplots titles.

Figures 3, 6 and 7: please enlarge the font used in the subplots

Figure 15: please enlarge the font used in the subplots and introduce units for Chl-a;

Figure 17: Upper and low left panels, xlabel(subtitle) brackets are missing at the end.

---

## Editor Comment (EC1) · Andrew Moore (Editor) · 21 Dec 2020

Dear Marion,

Both referees are of the opinion that your manuscript requires major revisions before it is acceptable for publication.

Please address in detail all of the comments addressed by each referee in your response and if you decide to revise your manuscript for further review.

[Figure]

Yours sincerely Andrew Moore

---

## Author Comment (AC1) · 19 Mar 2021

**OS-2020-100: Response to reviewer comments**

We would like to thank the two reviewers for their comprehensive and thoughtful comments.

In terms of the major comments we have taken what RC1 wrote about Section 3 on board and have restructured and in places reworded the section with a new figure to help give a better sense of the method and how it works.

In terms of the analysis in Section 4, we agree that the AMM7v8 *model* analysis and forecasts are not doing a very good job at present of forecasting chlorophyll concentrations, though the spatial method does extract more useful information in terms of where the major issues of poor performance lie, e.g. the timing error in the onset of the bloom. The AMM7v11 *model* analysis was not originally part of the study, and there were no AMM7v11 forecasts produced at the time, so these cannot be included.

Upon careful consideration, given the poor performance of AMM7v8 and the confusion of comparing three datasets, we agree with RC1, and have taken the recommendation of removing all aspects of the AMM7v8 analysis and forecasts (i.e. the three-way comparison) from the paper. This has meant a change of title and fairly substantive rewrite of Section 4, with many changes also needed in other sections to match. There have been other benefits. The original paper was excessively long. The removal of AMM7v8 has allowed for a shortening of the text and rationalising of the figures, providing a paper of more conventional length. This makes the paper much less confusing with a clearer message. It also means that many of the minor comments made by RC1 and RC2 are no longer relevant. Any comments related to deleted text are shown below with a .

We appreciate that the word "analysis" is often used to describe anything that is on a grid; therefore, the L4 ocean colour product is also an "analysis" in that sense. The text has been made a lot more generic, removing all mention of forecasts. The L4 product is no longer referred to as an analysis. This term is reserved for referring to AMM7v11 only.

RC2 requested some further background to the new method:

*To illustrate advantages of the presented verification methods (newly applied for Chl-a forecast evaluation), it would be nice to compare the proposed method with other metrics like, for instance, bias (classical), MAE(MAD), bloom phenological indices (Siegel et al. 2002, Soppa et al. 2016), or any methods previously used for AMM NWS Chl-a valuation (mentioned in lines 83 - 85) with respect to complication/simplicity, possible diagnostic (meaning) and conclusions drawn by the analysis.*

Such metrics have been used elsewhere in the literature to validate the AMM NWS Chl-a, and we have added further AMM NWS references on evaluation work to the introduction, and a summary of relevant findings, which serve to put the findings of this study in context as suggested.

*RC1: minor comments*

l 21: "whilst several forecast blooms did not materialise in the observations": This sounds like the data is to blame for not showing the bloom, maybe rephrase to something like: "while the model forecasts also showed blooms that do not appear in the observations".
The abstract has required a substantive rewrite given the change in focus of the paper

l 22: "Whilst the model...": Most of this has been said in the previous sentence, would

be more useful to move it one sentence up as an intro.

The abstract has required a substantive rewrite given the change in emphasis of the paper.

l 34: "double penalty effect": For readers not familiar with this term, it would be beneficial to describe this term a little better. Currently it sounds like not in the right place would be a single penalty and additionally not at the right time would be the double penalty.

The explanation has been strengthened to help the reader unfamiliar with the concept understand it better.

l 114: "the models at 7 km resolution cannot resolve the coasts": It is not entirely clear what exactly is meant here: are you referring to the coastal chlorophyll a dynamics?

It is a combination of the coastal dynamics, and the grid resolution which makes many of the processes sub-grid scale. This has been clarified in the text.

l 126: "is" -> "are"

Reworded.

l 133: It would be good to introduce the "Atlantic Margin Model" the first time "AMM" is used in l 117 or l 98.

Done. It has been added around line 87.

l 135: I have a little trouble understanding this: is "Day 4 for the period of 1 March-31 July 2019" simply March 4th or are multiple 4-day forecasts created?

l 136: Mention that both the analysis and the forecast are used in this study. The previous sentence is confusing to the reader, it appears to state that the forecast is hereafter referred to as AMM7v8, when later on AMM7v8 forecast and analysis are used.

All reference to AMM7v8 has been removed from the paper.

Fig 1: I would prefer to have the log scale included in the colorbar ticks ("10^{-3}" instead of "-3") rather than written out in the title. This would also eliminate potential confusion, as the values in (d) are not exponents, although the title may indicate that.

This has been changed and the colour bar of the Fig 1 top subpanels are now log scale, with actual values in the tick labels.

l 150: Use (a) and (b) instead of "left" and "right". Same for the next sentence.

This has been amended.

l 158: "can be comparable": That is a very imprecise statement, as a reader I am not sure what this is telling me. Are the observation errors of the same magnitude as the model bias?

This is not the case anymore when considering only AMM7v11.

l 166: The instrument to measure ocean colour is satellite-borne, the ocean colour is not. Maybe use "remotely-sensed" or "satellite-derived"? Also, I would not refer to colours as concentrations.

The vocabulary related to the satellite-derived chlorophyll concentration has been changed and clarified.

l 167: Out of curiosity, why is a different satellite product used for the comparison and

is there a significant difference between the two satellite products?

The product used for the assimilation is the L3 NRT product,  whereas the product used for the comparison is the gridded (L4) REP product (OCEANCOLOUR_ATL_CHL_L4_REP_OBSERVATIONS_009_091), it is available 6 months after NRT and benefits from the maximum level of post processing.

l 172: Is this referring to the coupling between physical and biological model components?

Yes. This has been clarified in the text.

l 173: Is the AMM7v11 data assimilation also based on 3Dvar?
Yes. This has been clarified in the text.

l 181: The "sequence of forecasts" is a bit confusing here. The next sentence refers to "a forecast". I know that data assimilation can create a sequence of forecasts but here I think it would be much easier for the reader to stick with "a forecast" throughout the description of MODE.
This section has been rewritten. We have removed the use of the word forecast and observed. The use of the word "sequence" here is justified because MTD cannot be used without having temporal sequence of some description.

l 184: The mention of a model-based analysis is not helpful to the reader here, and it also not used again in this paragraph. I would suggest to rephrase to something like: "in this context, one is typically a model forecast, the other an observed field, i.e. observations regridded to match the locations of the model grid." Later on it can be mentioned that it can also be used to compare two model fields.
This section has been rewritten as suggested.

l 189: "observed objects" -> "observed field"
We have removed all references to "observed" and "forecast" from the revised paper, so this sentence has been reworded in a more generic way.

l 198: "and is based on a disk": Is the convolution kernel really a (flat) disk? The "based on" is confusing here.
This section has been made simpler and clearer, omitting words and phrases which could confuse.

l 201: Maybe use "objects" again, as above, instead of "areas".
This has been done. See around line 222.

l 204: Are the observed fields not smoothed? In step 2 it sounds like both fields are smoothed.
This section has been rewritten substantially. Yes, both fields were smoothed.

l 212: This is unclear: the first sentence in (6) "which together are expressed as the socalled "interest" score" makes it seem like one interest score is computed summarizing the fit of all objects. In this sentence, "interest scores are computed for all" objects. Please rephrase.
This section has been significantly rewritten. Hopefully this aspect has been made clearer. See lines around line 228.

l 252: "minimum volume of 1000 grid squares": This should be "area" or is this applied

below the ocean surface? Why is it 1000 grid squares here and 10 above? Reading it
a few more times, it seems like this describes a "volume" in space-time, which needs
to be made more explicit. "1000 grid squares" is not a volume, is this 10 grid squares
times 100 time steps?
The thresholds are independent of each other in MODE and MTD. Here the 1000 grid squares does
refer to the accumulated number of grid squares over consecutive time slices. The 10 grid squares
refers to the MODE area threshold.  This has been clarified in the text, in and around lines 263.

l 255: I am guessing here that paired objects are those for which an equivalent was
found in the other field while all others are called single. And clusters are two or more
objects in one field, classified as belonging together. Please add a bit of description
here, to define these 4 terms.
This forms part of the rewrite of Section 3.1, which also now includes a schematic in a new Fig 2 to
provide visual aids to what these terms mean.

l 262: Using the log-transformation seems like a sensible choice for chlorophyll. It
would be good to know if the transformation was applied before smoothing the fields.
The fields were not transformed but the thresholds were derived from an equally spaced
progression in logarithmic space to ensure that the thresholds we applied followed the underlying
distribution of the Chl-$a$ values. Apologies if this seemed confusing.

l 266: "mg." -> "mg"
We believe that you would like us to remove the full stops in the units. Which we have done.

l 267: The units here would be "log_10(mg mˆ-3)". In my opinion, it would be clearest to
just provide the values in linear space, e.g.: "For this study, a range of thresholds were
applied to the log10-transformed chl-a fields, corresponding to chl-a concentrations
between 1.62 and 25 mg mˆ-3."
We have reworded the section to make it clearer what was done.

l 268: Why use 25 mg mˆ-3 here when the values of interest are in the range 3 - 5
(previous sentence)?
We were guided by the values that have been typically used in other studies. Over the NWS the
values do appear to be mostly in the range 3-5 mg.m$^{-3}$ but larger values are present, though in too
small numbers. This has been reflected in the text on line 276.

l 274: "This radius" -> "The 5 grid point radius"
This paragraph has been rewritten to reflect changes to the focus and emphasis of the paper to only
include AMM7v11.

l 276: Why so much discussion of different thresholds above when only a single one
will be used here? Or is "here" only referring to the sensitivity analysis? Please be
more specific.
A series of thresholds were considered as per line 267 in the original manuscript. We would never
have been able to discuss all of them, and in fact on the lowest thresholds provide sufficient sample
size for analysis. Line 276 was there to say that most of the results presented in the paper would
relate to the 2.5 mg.m$^{-3}$ threshold, which was the second lowest we considered. This has been
amended slightly near line 276 in the revised paper.

l 304: What is "this", maybe use "The effect of bias"

l 310: "would yield no useful information": One could argue that having no matched pairs contains the useful information that the model solution cannot be very good.
True. It provides a very stark outcome but in terms of trying to demonstrate what a feature-based method can provide, would not have been very helpful. It also doesn't totally capture the ability of the model to capture enhanced levels of Chl-a, which could still provide some useful information, with the caveat of knowing about the bias. We have added a sentence to reflect your comment, which is a valid one. Text has been added around line 329 in the revised paper.

l 315: Why is the L4 product referred to as an analysis here?
It is an analysis of sorts but the use of the term here, we agree, is sloppy. We have ensured that the text only refers to the AMM7v11 as an analysis and L4 is called a product.

l 324: "the two that clearly differ more dramatically from Fig 2": What is meant here is probably "the two fields that show the largest discrepancies in Fig 2".
Correct. This has been reworded, though the revised Fig 5 no longer shows quite the same discrepancies as AMM7v8 has been removed (panel (a)).

l 336: What about using different thresholds for the different fields?

This is essentially what quantile mapping does or the use of a fixed, but different, threshold derived from the dataset… which is what was referred to as the "seasonal" threshold. This is no longer relevant in the context of the revised paper.

339: "forecasts": Which forecast, only analysis solutions were considered thus far.

Fig 3: Please include the AMM7v8 distribution after the quantile mapping. And why not include the AMM7v11 distribution as well?
As the paper has been reduced in scope to only include the AMM7v11 analysis this figure now reflects only the L4 and AMM7v11 distributions. For completeness, it could have been interesting to plot the AMM7v8 quantile mapped distribution, the method did not use the season long distribution, but the daily mapped distributions.

l 356: "the observed threshold": Do you mean the threshold applied to the observations?
Yes, that is what was intended. We have refrained from using the word "observed" in the revised paper, choosing to refer to L4 as a "product" instead.

l 358: After a lot of reading the next paragraphs over and over to figure out what the thresholds are, this is the position where I lost track. Here, it needs to be stated that the "value that has the equivalent rank in the forecast distribution" is the "forecast threshold". They are currently not linked and initially I thought that the forecast threshold was the same as the (slightly misnamed) observed threshold.
This section has changed substantially to reflect the removal of AMMv8 forecasts from the paper, along with the use of variable thresholds. In doing so, the paper has become a lot more focused with less opportunity for confusion.

l 362: Based on this description it is not clear what this threshold is. It is also called "seasonal" here when Fig. 4 shows daily variations in a threshold. Are these the same thresholds, why is it called seasonal? If this is the threshold used to identify objects, please mention this explicitly.
We agree that this section was very confusing. With the decision to remove AMM7v8 this section is no longer relevant and has been removed. We have kept a revised version of Fig 4 to show the

distributions vary on a day-to-day basis and how this affects the AMM7v11 concentration value that is in the same centile of the distribution as the 2.5 mg,m$^{-3}$ L4 distribution.

l 365: Is this procedure applied to both v8 and v11?
No, it wasn't. But this is no longer relevant with the focus on v11.

l 374: "the latter": is this the L4 product? "the latter" is not really referencing anything at this point, please just use the product name.
This whole section has been reworded to reflect the change in storyline, hopefully making it clearer too.

l 375: Please explain better what this threshold means. It is impossible to understand this paragraph without knowing what the threshold signifies.
The discussion here has been reworded in light of the fact that the concept of quantile mapping has been removed. We felt it was useful to keep the concept of distributions on the daily time scale to provide as rounded a picture of the variations of the bias on a day-to-day basis, to explain the impact of using a fixed threshold for both datasets.

Fig. 4: It would be good to include a grid and reduce white space, the threshold never exceeds 6. I would further suggest to merge this figure with Fig 5.
Fig 4 has been recreated and Fig 5 has been removed as AMM7v8 is no longer discussed anywhere. Instead of a grid we have added a horizontal dotted line at 2.5 mg.m$^{-3}$ as a visual guide. We hope this is satisfactory.

l 390: "Error! Reference source not found." Something appeared to have gone wrong with the automated submission system.
Indeed! Not sure what happened. Sometimes the pdf conversion can also do this. Sorry we didn't spot this.

l 391: "using the built-in functionality in MODE as for Fig 4.": This is unclear, please rephrase.

As the quantile mapping has been removed from the paper, we have refrained from referring to the method in those terms. We have retained the figure in the text but referred to it in statistical terms rather than using any reference to MODE capability.

l 394: What is the forecast lead time?

l 405: "the forecasts are very active": What exactly does this mean, phytoplankton blooms?

l 406: "The latter object is not identified in the L4 ocean colour product." It is not clear what "object" this is referring to. I do not think this paragraph is very helpful to the reader at this point. So far the reader has only seen an example of what object identification should not look like (Fig. 2) and now here is a lengthy explanation of chlorophyll-a blooms leading to peaks in a threshold without showing an example of that these variable thresholds improve object identification.
This paragraph has been removed from the revised paper, and the preceding explanation of the methodology rewritten to be clearer

l 434: "this work was done without accounting for the concentration differences": Does this imply that the variable threshold from the previous section was not used? But it

seemed to be very import for successful identification.

Correct. We agree that, in hindsight, some of the elements in the submitted paper can appear contradictory. In the rewrite these sections, which refer to AMM7v8 have been removed as this part of the analysis was not done for AMM7v11, as it wasn't available at the time. We did use the smoothing radius findings for AMM7v11.

l 444: Are "quilt plots" the same as "quilt "difference" plots"? I would suggest to stick with one term.

l 444: Here are two nearly identical sentences two (short) paragraphs apart: "Figure 6 provides a selection of quilt plots derived from using the L4 ocean colour products and AMM7v8 analyses during July 2018, using one of the merging options which was tested." (l 435) and "In Figure 6 some quilt "difference" plots are shown to focus on the individual characteristics of the AMM7v8 analysis and the L4 ocean colour product based on a set of initial data that was available for July 2018."

l 451: How useful is this analysis if it is already clear from the previous section that a comparison at the same threshold is not sensible? The 2.5 mg mˆ-3 threshold can be established without considering the model output.

True. In the rewrite of the paper this section has been removed as we are no longer considering any aspects of AMM7v8.

l 475: Again, how valid are these results if different thresholds would be used for model and data? Would the conclusions about the smoothing radius hold?

This section has been removed as this analysis was not done using the AMM7v11 results, as they weren't available at the time.

Fig 6: The font is much too small.

This figure has been removed as it relates to AMM7v8.

l 489: Is this analysis done at the same threshold for data and model?

Yes, it is.

l 491: Is this "total area" of identified objects or ocean grid cells?

This is the total of the combined area of identified objects and has been clarified in the text.

l 534: The lead time has still not been properly explained.

l 545: So these percentiles are characterizing the chl-a distributions among different areas. How can they contain values below the 2.5 threshold applied to the observations? The "(in this case 2.5 mg mˆ-3)" makes it sound like the same threshold was applied to the two products but this seems to contradict earlier sentences. Assuming now that the threshold are different: From Fig. 3, we already know the distribution of chl values, what new information does Fig 9 give the reader? We know of the bias and have a rough idea of the distribution, and we also know that a higher threshold will likely be used for AMM7v8, which appears to be the main contributor to the differences between the distributions in Fig 9.

This figure has been removed as it relates to AMM7v8.

l 574: It would be nice for comparison to see good results for comparison. How would

Fig 10 look for a AMM7v11 to L4 comparison?

Fig 10 does show the AMM7v11 and L4 results, but not against each other. In the revised figure the AMM7v11 vs L4 comparison is the only comparison that is shown, which hopefully makes things clearer. Yes, it would be nice to show good results and these certainly are much better than for AMM7v8!

l 623: "AMM7v8 day 0 forecast to L4 and AMM7v11 (labeled AMM7v8 vs AMM7v11, and AMM7v8 vs L4)": The labels are either switched or cleverly selected to confuse the reader.

The three-way comparison has been removed so this figure has also been removed.

l 653: Is the spatial centroid the average location of the center of a series of identified objects? This needs to be explained.

Yes, it is. This has been clarified in the text.

*RC2 specific comments:*

Lines 15 – 16: I like that the discussed verification method allows not only identify bias but correct (mitigate its impact when carried other further analysis). How would it compare with regular bias correction?

The aspects of quantile mapping have been removed from the paper since it was not deemed necessary to evaluate AMM7v11 alone as the bias isn't as severe. How would quantile mapping compare to other bias correction methods? It depends on the data sets. Depends also on what you mean by regular. I bulk bias correction would affect all parts of the distribution in the same way and could therefore do some strange things in the tails of the distribution. Other forms of calibration-based method such as Kalman filtering would probably provide similar results but have other advantages in that they can extrapolate to values outside the distribution, if that is an issue.

Lines 526 – 530: Reads as this paragraph should not be at this place. Which figure is discussed? Figure 12?

This paragraph has been moved as it did appear to be in the wrong place. It has been to the discussion of the revised figure, though still Fig 9. The original Fig 9 has been removed entirely.

Lines 553 – 557: How would it compare with analysis of the frequency distribution (histogram)? Part 4.5 "Onset and evolution", I would suggest change the title to "Bloom onset and evolution". I am also curious how would the material and results presented in this subsection compare with simple bloom phenology analysis based a threshold method (Siegel et al. 2002, Racualt et al. 2012, Brody et al. 2013, Soppa et al. 2016).

Following the substantial rewrite this material has now been included in a new Section 4.4 titled "Incorporating the time dimension". Bloom phenology analysis based on threshold methods is not yet routinely used as a model validation metric, as applying a method consistently to both satellite and model data is, in practice, not straightforward. However, we have now referenced other studies which have assessed bloom timing in the NWS system (e.g. Skákala et al., 2020, https://doi.org/10.1029/2020JC016122), to provide some context for our findings.

Lines 772 – 774: "Blooms were said to occur when the observed concentration threshold exceeded 2.5 mg.m 3. Forecast thresholds for MODE were then relative to this value and varied from day-to-day. For MTD the seasonal equivalent threshold for the AMM7v8 forecasts was 6 mg.m 3". What can be in general concluded from this about AMM7v8 performance?

**Typos**
Line 99: delete on of two words "product".
Line 187: "*merged*", "*matched*" – is italic font urgently required. Or it is a format error?
Line 197: "*further*" - a format error?
Line 206: "*not*" - a format error?
Line 257: "*single simple*" - a format error?
Lines 217, 218, 220, 232, 233, 237, 238, 242: please check/confirm if italic font used for a number of words is required (?since refer to specific options? right?).
Italics was used for emphasis or to highlight specific key words relevant to MODE or MTD. This whole section has been rewritten and these are no longer relevant.

Lines 546 – 548: again, a format error – used square brackets, "*exceeding the threshold*", "*within-object*"
Lines 550 – 552: "*within-object*" - a format error?
This text was a description for Fig 9 which has been removed.

Line 641: "*and*" - a format error?

Line 652: *"time"* - a format error?
Italics removed.

**Figure quality**
Figure 2: please improve quality of the subplots titles.
The figure has been recreated with fewer panels to remove AMM7v8. This has improved the overall sizing etc. This is now Fig 5.

Figures 3, 6 and 7: please enlarge the font used in the subplots
Figure 3 has been recreated with larger axis labels and using the AMM7v11 dataset. Figs 6 and 7 have been removed with the rewrite.

Figure 15: please enlarge the font used in the subplots and introduce units for Chl-a;
Figure 17: Upper and low left panels, xlabel(subtitle) brackets are missing at the end.

---

## Author Response (AR2)

**Response to second review OS-2020-100 comments**

**Report #1**

The example shown in Fig. 5 is very helpful but could be expanded easily to be of even more use. To better understand the effect of the MODE parameters that were chosen, it would be nice to show the matched objects, for example by using the same color for objects that match, by connecting them with lines, or by adding numbered markers that also show the order in which they are matched. As a reader, I'd be interested, for example, if AMM7v11 objects 7 and 12 (off the coasts of Denmark and Germany) remain unmatched or are matched with the big L4 object 10. Adding information about the matching will help readers understand the results in the following sections 4.3 which focuses on the fraction of matched objects and problems when matching objects.

This is now Fig 6 and the colours have been adjusted so the matches are evident. Anything unmatched is coloured grey.

Another example of where an example could be expanded is Fig. 11: It would be nice to use the same colors as in Fig. 10 here, so that the bloom events can be related to each other. Lines could connect dots of the same colors, to further emphasize that they are associate with the same bloom event. Going one step further, Fig. 10 and 11 can be merged. And once colored, Fig. 11 appears to contain all the information contained in Fig. 10a, and 10a could be removed.

Now Figs 11 and 12, with Fig 12 merged to become part of Fig 11. Fig 11a very specifically refers to the duration each object existed for given the threshold to define it. Fig 11b now shows the areas (with a bug fix!).

Several smaller implementation aspects of the study remain vague or are not mentioned. I have pointed out several instances in my specific comments below, one example is the log-transformation of the chl-a field. The manuscript includes a somewhat lengthy description of the choice of thresholds and that they are equally spaced in log-space, just to translate them back into linear space and discard all but 3 of them for this study. Meanwhile, an important implementation aspect concerning the log-transformation -- if smoothing is performed in log-space (I presume so but I am not sure) -- is not mentioned at all.

Your queries about the thresholds has been addressed. Hopefully the discussion is now sufficiently clear. The data sets were not transformed but thresholds were chosen to be equally spaced in log-space, thus bisecting the data appropriately (given the underlying distribution).

**specific comments**

l 19: "By contrast the AMM7v11 analyses produces more bloom objects in deeper Atlantic waters, which are not detected by the satellite product.": Not if sure if this is intended but this formulation makes it sound like the model is correctly producing a chlorophyll bloom while the satellite (incorrectly) fails to detect it.

This does read ambiguously and has been reworded. In addition, the abstract has been shortened somewhat to conform to the 250 word limit.

l 72: "that is": It's not clear if "that" is referring to blooms or "oceanic primary production". I assume

it is the latter and would suggest to remove the "that is" and replace it with a ",".
This has been clarified, we have replaced with "," as suggested.

l 77: While there is definitely a link between Chl-a and phytoplankton biomass, many of the biogeochemical models often have separate variables for both phytoplankton biomass and Chl-a. So using model Chl-a instead of model phytoplankton biomass to infer phytoplankton biomass is not a common use case. But why mention biomass here at all, I would suggest to link Chl-a directly to blooms.
We have rephrased this sentence to say:

"Biogeochemical models coupled to physical models of the ocean provide simulations for the various parameters that characterise the evolution of a spring bloom, such as Chl-a concentration which can also be estimated from spaceborne ocean colour sensors (Antoine et al., 1996)."

l 84: Maybe add Mattern et al (2010) to the list here, which compares several neighbourhood-based methods for model-satellite Chl-a comparison.
We have added a reference to Mattern et al. (2010) as suggested.

l 117: Maybe add that the model grid is used for this purpose here.

This has been added.

l 125: "chlorophyll": Shouldn't this also be Chl-a?

This has been changed.

l 158: I would suggest to add one sentence about the difference between this product and the L4 product used in the comparison. Even just to mention that this is not the same dataset later used in the comparison.
We have added the following sentence:

"The L3 product is based on two of the same three ocean colours sensors used in the L4 product described in Section 2.1, but with different processing and no gap-filling."

Fig. 1: While "L4 observations" is no longer used in the text, it is still present in the figure and its caption. I am not against using the term but for consistency, the authors may want to change it.
This was changed in figure and caption

Fig. 1: Negative exponents on the color bar are not fully in the superscript.
Color bar corrected in a revised figure.

l 212: Use "Fig. 2a" instead of "a", I was at first confused what it was referring to.

This has been changed. Fig 2 has been added to the (b) as well for clarity.

l 216: "Only the pairing with the highest score is analysed further": In this particular example, or in general? What would happen if the 3-2 pairing scored 0.75 in Fig. 2b? I assume it would also become a match but the text suggests it would not.

No, only the pairing with highest score is analysed as a matched pair, though the results are stored for all the matches that are performed. Each possible combination of objects is scored. Whether a particular match is then retained depends on whether the score is above the threshold (0.7).

Fig. 2: Nice example but it would be useful to more explicitly mention that 1 and 1 are matched. Currently only the bold text suggests it.

Good point. We have added this to the text.

Fig. 2: The 1km distance bar should be moved closer to the objects in 2a.

Done.

l 236: In the answer to my previous comment, the authors mention that the observations are smoothed as well. This should be made explicit here.

This has been clarified.

l 278: When mentioning the threshold of 1000, it would also be good to mention the time resolution that is used here. Is a distance of one unit in time treated equally to one unit in space?

In this instance the volume is the sum of the grid squares identified in each time slice of the identified space-time object. This has been added to the text.

l 290: Maybe write "to _only_ capture events of interest" ("only" added by me) to emphasize why that would mean increasing the threshold.

"only" has been added.

l 293: "though higher values are present": In the data, in the L4 product, or both?

This is somewhat ambiguous and has been clarified.

l 296: "In the paper": Here? I would suggest to rephrase to "here", "in this study" or "in this paper".

The phrase has been replace with "Here…."

l 304: Why is a count of 30 an important limit?

These methods are intended to be inherently heuristic in that they attempt to mimic what a human would do. This number was arrived at through some pragmatic eyeballing. Beyond that number my brain could not have managed to make much sense of what to do with the identified objects!

l 304 "Furthermore, the smoothing applied needs to be reduced with increasing concentration thresholds because objects become smaller and are less frequent.": Here the logic appears to be

backwards: smoothing happens before applying the threshold. So, increased smoothing decreases the concentration, requiring a lower threshold.

Indeed, you want to apply less smoothing for the higher thresholds to avoid destroying the peak values before identifying objects (noting that the raw field information is put back into the objects after object identification). A figure similar to the one below (I think) was originally included in the manuscript, which showed the inverse relationship between the number of objects as a function of smoothing radius and concentration. See the figure below… these were results for the 2018 season sensitivity analysis but highlight the principle. It also makes it very clear that setting the threshold too low isn't a very good use of this methodology. You want to identify distinct features or objects.

[Figure]

Quilt plot for sensitivity analysis: number of objects identified as a function of convolution radius R (number of grid squares) and threshold T (mg/m³)

l 308: Remove "too".

Done.

l 308: "objects may be spurious": It is not really clear what this mean in this context. Wouldn't increasing the threshold remove objects that are less relevant?

Not using a smoothing radius along with an inappropriate area threshold for object identification could result in objects being identified which are actually not meant to be objects. This is especially true for gridded observation fields, such as rainfall estimates from radar where the presence of ground clutter can still remain, even with reasonably good QC. You do not want to include objects like this in the analysis. The wording has been strengthened in the text.

l 309: "AMM7v11 analyses are on a ~7 km grid." This has been mentioned before.

Fig. 3, 4, 6, 7, 11: It would be useful to show grid lines in the panels

In many of these plots there are lot of lines. What is now Fig 12 has been removed and integrated with Fig 11. We have opted to include some (feint) reference lines where necessary, e.g. Fig 5.

l 332: "centile": This is the short form and less commonly used word for "percentile". Especially since percentages are never used here, I would recommend to use the more general and likely better known term "quantile" instead.

Quantile has been applied throughout.

l 332: "less than equal": Shouldn't this be "equal"?

We believe this is correct.

l 333: "corresponding AMM7v11 centile": This needs to be described a bit better, is the "corresponding" centile/quantile the quantile of the same value or the quantile that is associated with the same chl-a concentration?

The explanation has been further strengthened with specific reference to Fig 4.

l 350: Maybe a better example (rather than a constant value) would be the presence of high chl-a features in the right place, i.e. "A lack of objects suggests the presence of a model bias but it does not provide any sense of whether the model is producing enhanced Chl-a (albeit below the chosen threshold) at the right location, or not."

The fundamental premise of this method is to take away the need for the forecast (or analysis) to produce the feature in the right place at the right time, as this is a specific issue with high-resolution models. The method of equalising quantiles (or quantile mapping) works on the premise that one distribution is fixed and the other is not, and in doing so the bias is removed. This is why we had included the work on quantile mapping in earlier revisions, but this seemed to confuse rather than help in explaining the method and so the fixed threshold results have been shown here instead. Indeed, the presence of a bias, could lead to the scenario you describe. Hence, why understanding what impact the bias *might* have, is really fundamental to any threshold-based methodology. Here, we are relatively safe as, though there is a bias, it isn't prohibitive in the way you describe.

l 360: "In this instance, though there is bias, it did not prevent the identification of objects in either fields to the extent where the results did not reflect the potential for the analyses to provide features which could be matched, paired and compared.": This sentence needs to be rephrased and simplified.

We have attempted to do so.

l 415: "From an interpretation perspective": It's not clear what this means, I would recommend not using ambiguous phrases like this.

We have tried to clarify the text.

Fig. 7: Because the description explicitly mentions the total number of objects and because the lines often overlap, it may make for a better plot to show the total number of objects and the number of matched objects, instead of matched and unmatched.

The figure has been replaced to reflect the total vs matched number of objects and the text amended.

l 441: Where are the "South West Approaches"? I'd suggest to either describe the location in different terms or mark it on the map.

This has been rectified with the inclusion of new Fig. 2, and is referred to in the text.

l 441: "There are areas, for example in the South West Approaches, where there appears to be a good level of consistency.": But this is for identified and not matched objects, correct? How about similar maps that show the probability of the grid cell being part of a match given that is is part of an identified object?

Indeed, this would be very interesting to do, and is a great idea. The gridded output is not currently sophisticated enough to allow that to be done with any ease. Also, for this dataset the probabilities would be very low. The maps here provide more of a climatological view of the season.

Fig. 10: Does the same color for AMM7v11 and L4 indicate that the objects matched?

This is now Fig 11. No, the colours here denote the progression of time… approximately the same hues give a sense of the temporal proximity of the objects, but as (c) shows even if objects exist at the same time, they may not be geographically close.

l 566: Are these 26 days solely based on the first bloom object (l 501)?

This is the difference in dates between the first identified bloom objects in both datasets. This has been clarified more in the text. Now near line 543.

l 587: "MODE results suggest that the AMM7v11 bloom objects are larger than those in the L4 product.": That appears a bit surprising given the results in Fig. 5 showing large L4 objects, the higher chl-a concentration in L4 (see distribution), and the result that AMM7v11 objects are numerous but small.

This is now Fig 6, which is only one daily snapshot. The area panel in Fig 11(b) shows the area differences more clearly.

**Report #2**

My main concern still remains: to illustrate advantages of the presented verification methods (newly applied for Chl-a forecast evaluation), it would be nice to compare the proposed method with other metrics like, for instance, bias (classical), MAE(MAD), bloom phenological indices (Siegel et al. 2002,

Soppa et al.2016), or any methods previously used for AMM NWS Chl-a valuation with respect to complication/simplicity, possible diagnostic (meaning) and conclusions drawn by the analysis given the particular (addressed in the paper) application case/data. The study would benefit from such a comparison that should not not take lots of time.

While the focus of the paper is on introducing and exploring new methods for evaluating Chl-a, we agree that some comparison to classical metrics may be helpful for the reader. We have therefore included median bias and median absolute difference as per your suggestion, and additionally Pearson correlation coefficient. These are now shown in Tables 2 and 3 and Fig. 2, with accompanying discussion in Section 4.1. These metrics are the ones used in the CMEMS Quality Information Document for the model product (McEwan et al., 2021, https://catalogue.marine.copernicus.eu/documents/QUID/CMEMS-NWS-QUID-004-002.pdf), and so represent the routine metrics which the novel method introduced in this study would aim to complement.

We have not included bloom phenological indices as these are not commonly used for model validation. Siegel et al. (2002) and Soppa et al. (2016) are observation-based process studies for instance rather than model validation papers, and we are only aware of phenological indices being used to validate 1D models at in situ time series stations (e.g. Anugerahanti et al., 2018, https://doi.org/10.5194/bg-15-6685-2018), rather than 3D forecast models. Appropriate application of such indices to validating daily model outputs would require optimising e.g. smoothing and time-averaging procedures, which is outside the scope of the present study.

We have added the following text to the discussion section:

"Using MODE and MTD clearly gives extra information not obtained from traditional verification metrics that are more routinely used (McEwan et al., 2021). An alternative approach to assessing the representation of phytoplankton blooms might be to use phenological indices (Siegel et al., 2002; Soppa et al., 2016), which measure the day of the year on which Chl-$a$ concentration first crosses a threshold based on the median concentration. Phenological indices have been used in observation process studies (Racault et al., 2012), but very rarely for model verification, and then only in 1D (Anugerahanti et al., 2018). One reason for this is that daily model Chl-$a$ will frequently cross such a threshold throughout the bloom season, meaning temporal smoothing and other processing (Cole et al., 2012) would be required, which is not straightforward to apply consistently. Objective methods such as MODE and MTD, which consider individual bloom objects throughout the season, rather than assuming a single spring bloom will occur at each location, bypass these difficulties."

---

## Author Response (AR3)

**OS-2020-100.R2 final review responses**

We would like to thank both reviewers for their thoughtful and detailed review of the paper. You may be aware that the project this work is based on concluded in March 2020. Getting this paper completed has taken a monumental effort from all co-authors who have made this happen without any specific project hours since the project ended. We did feel, at the end of the project, that this work should be published, despite knowing that it would be challenging to achieve. It hasn't been easy but I am proud that we have managed to achieve that.

==============================================================================

*General comments*

I like the addition of the "traditional" statistics but it would have been nice to incorporate them a bit better with the rest of the manuscript, that is, bringing them up when the MODE/MDT results are described and discussed.

The final discussion section has been prefaced by adding a sentence which hopefully provides the missing link between the results in Table 1 and the related discussion.

*Specific comments*

l 17: Add that the verification methods helped to reveal the following model results.

The text has been adjusted. To respect the 250-word count limit for abstracts the text has had to be adjusted slightly elsewhere.

l 117: Maybe mention straight away that this model grid is coarser than the data grid and that the data is interpolated onto the model grid

Done.

l 219: "For the example blue object 1" -> "In the example in Figure 2(b), blue object 1"

Done.

Fig. 2: Should be "centroid distance" and not "centroid difference".

This is definitely the centroid difference. Each of the objects have a centroid location, and the arrow denotes the difference in these two locations.

l 304: I would suggest to introduce the thresholds one sentence before they are used, i.e.: "three thresholds of 2.5, 4 and 6.3 mg m-3 were selected to be equally spaced in logarithmic space"

Done.

l 345: "but for higher thresholds (e.g. 6.3 mg m-3 )": Since only one is used here: "but for the higher 6.3 mg m-3 threshold"

Done.

l 352: "Statistics for matched pairs": I would remove the word "matched" here to not confuse the

reader, as no matching in the MODE/MTD sense was performed. Or if it was, this needs to be elaborated on.

Upon reflection, leaving the word "pairs" behind may still be confusing. It may be expedient to take out the entire phrase "matched pairs", as it has a specific meaning within the context of this paper.

l 404: "(as seen in Figure 4 Figure 5)": Should just be Figure 4, I assume, or an "and" is missing.

The "and" has been added! Figure referencing didn't work as it should!

l 409: "the quantile mapping functionality within MODE": Is there a reference you could add for a paper that describes this technique in more detail?

Not in the MODE context. In the weather verification community, it is well known method for removing the bias. It tends to exist implicitly in papers, i.e. I can't recall it being flagged up as needing greater explanation. That is why moving concepts to other communities can be so useful. Things that are taken for granted in one, may not be in another. I could add Mittermaier et al. (2013) which references a different spatial methodology, but I am not sure that it would not confuse, and in truth, it does not really explain the methodology in any detail. This paper is likely to be the most descriptive one out there!

Mittermaier, M.P., N. Roberts and S.A. Thompson, 2013: A long-term assessment of precipitation forecast skill using the Fractions Skill Score. *Meteorol. Apps*, **20**, 176-186.

Fig. 6: This is a really nice plot showing the results of MODE. I notice a few new objects in the western part of the domain, compared to the previous version of the manuscript, which I assume were dropped accidentally from the old version of the figure.

Not dropped accidentally but the previous version was a more zoomed in version.

l 436: "suggest" -> "indicate"

Done.

l 530: Here and in the following line, the 25th percentile becomes the 0.25 quantile.

Sorry, but we can't find this.

Fig. 11: Again, nice figure! Minor complaint: It is difficult to distinguish some of the colors, like the shades of purple, is there something that could be done to better match the temporal with the spatial location?

We've changed the colour palette so there is no recurrence of the purple… sadly the early times do appear quite purple for some reason. They should look bluer, but didn't. A slight tweak in the base colour appears to improve this. The colour palette change will hopefully make it clearer and should make the cross-referencing less confusing. The graphics are not intended to provide perfect cross-referencing but are there to show the progression of the season from south to north, for example.

*Specific comments*

L94-96 the sentence started with "A full traditional" could be slightly rephrased given the following sentence or removed.

The sentence has been removed.

L108-110: Currently sounds abrupt. It would be nice to emphasize once again the aim of the study.
A sentence has been added to try to provide a smoother transition to the layout of the paper.

L298-299: please provide a reference on using log10 threshold, or please rephrase

The Campbell (1995) reference documents the fact that Chl-a follows a lognormal distribution. The reference for the use of log10 thresholds was missing and has been added.

It is always statistically advisable to consider the underlying statistical distribution when doing anything categorical, like using thresholds.

Consider below the observed Chl-a values for 2018 season, plotted untransformed (left) and transformed (right). The transformed version will better reflect the upper tail of the distribution. Using thresholds that mimic a transformation will have much the same effect as transforming the data.

[Figure]

```
> qobs # untransformed quantiles
          0%          10%          20%          30%          40%          50%          60%
4.899689e-05  0.06169716   0.1233453   0.1849935   0.2466417   0.3082898   0.5042315
         70%          80%          90%         100%
   0.7001732   0.8961149    1.092057    1.287998
> lqobs # transformed quantiles
          0%          10%          20%          30%          40%          50%          60%
   -4.309831    -3.550073    -2.790315    -2.030557    -1.270799   -0.5110408   -0.3868496
         70%          80%          90%         100%
  -0.2626584   -0.1384672  -0.01427595    0.1099153
```

Instead of using a fixed width sequence of thresholds for the untransformed values, which are heavily skewed, one can mimic the transformation by defining a set of thresholds which are equally spaced in logarithmic space, but are not equally spaced in reality, such that:

```
> 10^seq(0,1,0.2) # sequence of equally spaced thresholds in log space transformed
back to real space

1.000000  1.584893  2.511886  3.981072  6.309573 10.000000
```

Which is rounded to 2.5, 4 and 6.3 for this study.

L304: "were not log-transformed" instead of "were not transformed"
Done.

L306: could it be explained clearer?

See the response for l298-299. The text has been amended slightly which will hopefully make this clearer.

L321: Should it be "pushing the concentration threshold too low" instead of "pushing the concentration threshold too high"? otherwise sounds contradictory.
Spurious spikes in observed concentrations could be picked up when pushing the threshold too high, which are probably not real. When the threshold is too low, there are just too many objects to make sense of.

L322: is something missing in the expression in the brackets?
No. The e.g. has been replaced with a "for example" to help.

L329-330: is there also a 4 km resolution L4 product?
A global product is available at 4 km resolution, but the regional product designed for use in the Atlantic is only available at 1 km resolution.

L635: the following references on bloom phenology analysis in modelling could be provided: Ardyna et al. (2017), Hague & Vichi (2018), Pefanis (2021), Rohr et al. (2017), Song et al. (2010).
Thank you for highlighting these papers. We have added references to Pefanis (2021) and Hague and Vichi (2018), see the revised text below. We have omitted the other three, as they are process studies focussing on chlorophyll maxima or rates, rather than verification studies using the type of phenological indices discussed here.

"Phenological indices have been used in observation and model-based process studies (e.g. Racault et al., 2012; Pefanis, 2021), but rarely for model verification, and then usually in 1D (Anugerahanti et al., 2018) or at low temporal resolution (Hague and Vichi, 2018)."

*Typos/misprints*

L73: citation format: remove the bracket before Shutler;

Done

L84: please correct citation formatting: currently there are double brackets;

Done

L209: mind the font used for "merged" and "one";

As stated previously the italics is deliberate, used for emphasis.

L211: mind the font used for "matched";

As above, the italics is deliberate.

L404:"in Figure 4Figure 5" – please correct;

Done.

L529: mind the font used for "and";

As previously, the use of italics is deliberate.

L566: should it be "in July" instead of "into July"?

"into" is correct